# Regional Differences, Dynamic Evolution and Convergence of Public Health Level in China

**DOI:** 10.3390/healthcare11101459

**Published:** 2023-05-17

**Authors:** Jixia Li, Mengzhi Xu, Tengfei Liu, Can Zhang

**Affiliations:** 1School of Government, Beijing Normal University, Beijing 100875, China; 2School of Business Administration, The Open University of China, Beijing 100039, China

**Keywords:** public health, regional differences, dynamic evolution, spatial convergence

## Abstract

People’s health is a necessary condition for the country’s prosperity. Under the background of the COVID-19 pandemic and frequent natural disasters, exploring the spatial and temporal distribution, regional differences and convergence of China’s provincial public health level is of great significance to promoting the coordinated development of China’s regional public health and achieving the strategic goal of a “healthy China”. Based on China’s provincial panel data from 2009 to 2020, this paper constructs an evaluation index system for China’s public health level from five dimensions: the popularization of a healthy life, optimization of health services, improvement of health insurance, construction of a healthy environment, and development of a health industry. In this paper, the entropy method, Dagum Gini coefficient, Kernel density function and spatial econometric model are used to analyze the spatiotemporal distribution, regional differences, dynamic evolution and convergence of China’s public health level since the new medical reform. The study found that, first, China’s public health level is generally low, structural contradictions are prominent and the construction of a healthy environment has become a shortcoming hindering the improvement of China’s public health level since the new medical reform. The public health level of the four major regions showed a spatial distribution pattern of “high in the eastern, low in the northeastern, central and western” areas. Second, the overall Gini coefficient of China’s public health level showed a “V-shaped” trend of first decreasing and then rising, but the overall decrease was greater than the increase, among which the regional difference was the main source of regional differences in China’s public health level, but its contribution rate showed a downward trend. Third, except for the basic maintenance of a healthy environment, the Kernel density curves of China’s public health level and its sub-dimensions have shifted to the right to a certain extent, and there is no polarization phenomenon. Finally, the level of public health in China has a significant spatial correlation. Except for the northeast region, the growth rate of low-level public health provinces in China and the other three major regions is higher than that of high-level public health provinces, showing a certain convergence trend. In addition, the impact of economic development, financial pressure, and urbanization on the convergence of public health levels in the four major regions is significantly heterogeneous.

## 1. Introduction

In March 2009, the issuance of the “Opinions on Deepening the Reform of the Medical and Health System” marked the official beginning of a new round of medical and health system reform in China (hereinafter referred to as the “New Medical Reform”). After more than ten years of exploration and practice, by the end of 2021, the government had spent CNY 1.9142 trillion on health care, CNY 1.031 million medical and health institutions, CNY 9.450 million beds in medical and health institutions and CNY 13.985 million medical and health personnel, an increase of 4.792 times, 1.125 times, 2.140 times and 1.797 times, respectively, compared with 2009 [1]. With the joint efforts of the central government and medical and health units at all levels, the “New Medical Reform” has made remarkable achievements and contributed “Chinese wisdom” to the development of global medical and health undertakings [2]. Many data indicate that China has made certain breakthroughs in the fields of medical care and medical insurance. However, in 2016, China’s archived data showed that 5.53 million households and 7.34 million poor households were impoverished due to illness and returned to poverty, accounting for 44% of the total poor population, and major diseases, chronic diseases and endemic diseases became the main causes of poverty and return to poverty, seriously hindering the sustainable development of China’s economy and society [3]. According to the 2021 China Statistical Yearbook, the gap in life expectancy between different provinces in China can reach 10.36 years, and the difference in maternal mortality is more than 12 times [4]. At the current stage, the significant differences in health levels among different regions have become a constraint on the sharing of development achievements by all people. The Chinese government aims to solve the problems of low-quality medical and health services, imbalance in resource allocation, and significant differences in public health levels. In October 2016, the “Healthy China 2030” Planning Outline identified “promoting the equalization of basic public services in the field of health, narrowing the differences in basic health services and health levels between urban and rural areas, regions, and populations, achieving universal health coverage, and promoting social equity” as one of its four principles (health priority, reform and innovation, scientific development and fairness and justice) [5]. In April 2020, the National Health Plan for the 14th Five-Year Plan reaffirmed the basic principle of “accelerating the improvement of fairness and accessibility of basic medical and health services, and reducing the differences in resource allocation, service capacity, and health level between urban and rural areas, regions, and populations” [6].

In recent years, with the intensification of climate change and the increasing seriousness of environmental pollution, natural disasters and infectious diseases have emerged one after another [7]. The COVID-19 epidemic at the end of 2019 and the extreme rainstorm in Zhengzhou on 20 July 2021 have brought great challenges to the safety of life and property of the Chinese people and economic and social development. Under the new historical background of frequent natural disasters, the spread of infectious diseases and the rise of “healthy China” construction as a national strategy, how to scientifically measure China’s provincial public health level? What are the spatial and temporal evolution characteristics of China’s provincial public health level? What are the regional differences in the level of public health at the provincial level in China and the main sources? What is the dynamic evolution of China’s provincial public health level? Is there a certain convergence trend? The scientific and comprehensive answers to these questions can more systematically and objectively understand the development status, evolution rule, regional differences and convergence characteristics of China’s provincial public health, and provide certain experience support for promoting the coordinated development of regional public health, in-depth implementation of the “Healthy China” strategy and China’s participation in global health governance. Compared with existing research, this paper mainly expands public health-related research from the following aspects: in terms of research content, this study breaks through previous studies that only focus on the spatiotemporal distribution and influencing factors of public health at a single level. Using the Dagum Gini coefficient, Kernel density function and spatial econometric models, it explores the basic laws of China’s public health level from multiple levels, such as regional differences, dynamic evolution and convergence. In terms of research methods, this study considered the spatial spillover effect of public health levels, corrected the strict assumptions of traditional econometric models and ensured the accuracy of the calculation results. In terms of indicator design, this study combines the actual situation in China with the guiding ideology and strategic goals of the “Healthy China 2030” Plan Outline and constructs comprehensive indicators, including healthy living, health services, health security, health environment and health industry. This enriches the indicator system for public health evaluation and corrects the deviation of previous single indicator measurements.

## 2. Literature Review and Research Hypotheses

The COVID-19 epidemic that broke out at the end of 2019 has severely damaged the production and life of people around the world, causing hundreds of millions of people in developing countries to fall back into poverty and further exacerbating public health inequalities [8,9]. In fact, the origin of health inequality can be traced back to the first “health inequality” research group established by the British government in 1977. The following year (1978), the Almaty Declaration adopted at the International Conference on Primary Health Care clearly stated that fully achieving health for all and narrowing the health gap between developing and developed countries are the primary tasks for building a new international economic order [10]. In 1980, the Health Inequalities Group formally presented a report to the British Parliament documenting health inequalities in society and arguing that differences in health levels between different classes were mainly due to their economic and social circumstances. The report’s release immediately sparked a worldwide boom in research on health inequalities or disparities [11]. In 2000, five of the eight core Millennium Development Goals (MDGs) adopted at the United Nations Summit addressed public health and health equity. In the more than a decade since the release of the Millennium Development Goals, with the joint efforts of governments, the number of child deaths and maternal mortality rates around the world have dropped significantly [12], and the Millennium Development Goals have been basically achieved, but it is undeniable that the improvement of the average value of various health indicators is closely related to the fairness of people’s public health sacrificed by some countries. During this period, some scholars have transferred the concepts of income inequality and opportunity inequality to health inequality and have made preliminary interpretations of their connotations [13,14,15,16]. Scholars in different countries have also tried to measure health inequalities using single indicators such as mortality, life expectancy and self-rated health index and found that there are significant health inequalities in different countries such as New Zealand, South Korea and Chile [17,18,19,20]. With the complexity of health connotation, some scholars have begun to use input–output indicators to evaluate the allocation efficiency of medical and health resources at the hospital or regional level [21,22,23], with a view to providing empirical support for a more scientific grasp of regional public health levels. 

As research continues, scholars begin to explore the root causes behind health inequalities and the causes of health disparities that affect public health. Integrating the research results of multiple mathematicians, the factors that affect public health levels and lead to health disparities can be classified into four categories: natural environment, economic development, social life and public policy. (1) In terms of environmental factors, early studies found that the health level of residents in different regions has significant “local” characteristics [24,25], and Mariana Arcaya et al. used 1999 data on county-level life expectancy in the United States to prove the impact of geographical factors on health level [26]. In addition to geographical factors, some scholars have found that the ecological environment is also an important reason for affecting the health level of residents [7]. (2) In terms of economic development factors, another important factor in regional health level differences is economic development, and Bendavid E et al. found that economic development level is inversely correlated with child mortality from 2002 to 2012 through research on child mortality in developing countries [27]. (3) In terms of social life factors, Amador C et al. proved that lifestyle and social and environmental factors are the root causes of individual health differences through the recording of the genome-wide genetic information, lifestyle and economic and social environment of 11,000 obese people in Scotland [28]. Patrick Hoang-Vu Eozenou et al. found that social health service coverage also has a significant impact on individual health [9]. In addition, Zhaoxue Ci used data from the China Health and Nutrition Survey (CHNS) to explore the impact of income on health [29]. (4) In terms of public policy, in addition to natural, economic and social factors, some scholars have found that public policy will also have an important impact on regional public health, mainly involving trade policy, tobacco and alcohol policy and urban infrastructure policy [30,31,32,33]. 

Since the 1980s, with the widening of social disparities and the improvement of people’s health needs, the issue of health equity has begun to receive widespread attention in China. Early scholars mainly used data envelopment analysis (DEA) to measure the efficiency level of medical and health expenditure [34,35], and later some scholars began to calculate the efficiency of medical and health resource allocation in different regions such as urban and rural areas [36,37]. However, more and more scholars have found that only exploring the efficiency level of medical and health resource allocation cannot reflect the composite concept of public health. According to the United Nations Millennium Development Goals, Zhao Xueyan et al. selected the neonatal mortality rate, maternal mortality rate and infectious disease morbidity and used the entropy value method to measure the health level of residents [38]. Based on “Healthy China 2030”, Yang Fan et al. built a comprehensive evaluation index system for health level, explored the regional differences in China’s health construction level and found that China’s public health level showed an unbalanced trend toward a high east and low west [39]. 

In summary, existing studies have laid a solid theoretical foundation for the regional differences, dynamic evolution and convergent evolution of China’s provincial public health level under the background of the new medical reform. However, on the one hand, the above studies mostly use a single indicator or focus on the efficiency of input and output to measure the difference in public health levels, and the indicators are mostly absolute data, and ratio data and comprehensive assessment are less involved, which cannot truly reflect the current development situation in the new era of increasingly rich public health connotations. On the other hand, most of the existing studies have stayed at the level of depicting the evolution of time and space without further exploring their regional differences and dynamic evolution and failing to fully consider the spatial spillover effect of public health. In addition, existing indicators cannot fully integrate China’s specific national conditions to reflect the overall requirements of the “Healthy China 2030” Planning Outline for China’s public health. Therefore, the main contributions of this paper are as follows: (1) starting from the general requirements of China’s public health put forward in the “Healthy China 2030” Planning Outline, drawing on the reasonable content of public health level evaluation in existing research and building China’s provincial public health level evaluation index system from five aspects: “popularizing healthy life, optimizing health services, improving health insurance, constructing a healthy environment, and developing health industry”. (2) Dagum Gini coefficient and kernel density estimation method were used to characterize the regional differences and dynamic distribution of provincial public health levels in China since the new medical reform. (3) Establish a spatial econometric model to explore the convergence characteristics and influencing factors of China’s provincial public health level since the new medical reform so as to provide certain quantitative support for improving citizens’ health level and promoting the coordinated development of regional public health. Explore specific reasons: China has a vast territory with uneven resource endowments and economic and social development in various regions, as well as significant regional differences in public health levels. The eastern region has been affected by the reform and opening-up policy, and the process of industrialization and urbanization has rapidly advanced, attracting a large number of medical and health talents to gather here, establishing a relatively complete medical and health system, and the overall level of regional public health is relatively high. The central and western regions are located inland, with scarce resources and limited information. The level of economic and social development is not high, and they have long faced the problem of “difficult employment and retention”. Medical and health resources are scarce, the public health system is not sound, and the overall level of public health in the region is relatively low. Based on the above analysis, hypothesis 1 of this article is proposed: there are significant regional differences in China’s public health level, and inter-regional differences are the main source.

In addition, according to the first law of geography, everything within a spatial range is related, and if the distance is different, the interaction between the two also varies significantly. This spatial interaction is also understood as a spatial effect. This spatial effect can be divided into spatial dependence and spatial heterogeneity. Spatial dependence mainly refers to the fact that individuals in space are not independent of each other but rather interconnected, which is mainly caused by the spillover of factors, technologies and policies between regions. Spatial heterogeneity is due to different geographical locations and natural resource conditions, leading to certain differences between regions, such as coastal and inland, southern and northern as well as eastern and western regions. However, existing studies on the evaluation of public health levels often use traditional econometric models, which assume that individuals exist completely independently in space, do not comply with the first law of geography, and there is a certain computational bias. This study starts from the actual situation in China, based on the guiding ideology and strategic goals of the “Healthy China 2030” Plan issued by the Central Committee of the Communist Party of China and the State Council, and draws on the reasonable parts of existing research, attempting to construct a Chinese public health evaluation index system from five dimensions: popularizing healthy life, optimizing health services, improving health insurance, constructing a healthy environment and developing health industry. This evaluation index includes various levels of economy, society and ecology and involves the flow of various resource elements; therefore, there is a significant spatial correlation. In addition, the convergence theory of neoclassical economics suggests that under the condition of diminishing marginal utility of capital in various regions, the growth rate of economically underdeveloped regions is higher than that of economically developed regions. With the promotion of technology, this gap continues to decrease over time, and the economic development level of each region is ultimately in a balanced state. Therefore, the convergence theory can also be applied to the development process of public health in China, where low-level provinces of public health will gradually narrow the gap with high-level provinces of public health under the combined effect of technology and policies, presenting a convergence characteristic. Based on the above analysis, hypothesis 2 of this paper is proposed: there is a significant spatial correlation in China’s public health level, and it shows a certain convergence trend over time.

## 3. Methodology

### 3.1. Indicator System

Health is a state of complete physical, mental and social well-being and not merely the absence of disease or infirmity [40]. Good health is the basis for achieving all-round personal development and national prosperity, especially in such a post-epidemic era with frequent natural disasters and the spread of infectious diseases; understanding China’s provincial public health development level and its regional differences is of great practical significance for formulating scientific and effective medical and health policies. This study starts from China’s actual situation, based on the guiding ideology and strategic objectives of the “Healthy China 2030” Planning Outline issued by the Central Committee of the Communist Party of China and the State Council. It draws on the reasonable part of the existing research and attempts to construct China’s public health evaluation index system from five dimensions: the popularization level of a healthy life, the optimization level of health services, the improvement level of health security, the development level of a healthy environment, and the development level of a health industry; the specific indicators are shown in Table 1.

(1) Popularization level of healthy living. Healthy living is an important foundation for preventing disease and maintaining vitality. The primary task of improving China’s provincial public health level is to improve the foundation of physical fitness and form a healthy lifestyle. Among them, basic education for healthy life mainly reflects the average life expectancy, average education level and mortality rate of the region. Healthy lifestyles are mainly measured by per capita cultural and tourism costs, the number of public health activities per 10,000 people and the number of health and hygiene training per 10,000 people. 

(2) Optimization level of health services. Health services are a key link in promoting people’s healthy lives. Improving the level of provincial health services in China requires not only a large amount of medical and health resources investment but also to meet the medical and health needs of the people. Medical and health investment is reflected in three aspects: manpower investment, capital investment and material investment. The specific indicators are the number of health personnel per 10,000 population, per capita medical and health care expenditure and the number of beds in medical and health institutions per 10,000 population; medical and health needs are mainly measured by outcome indicators such as average hospital stay, maternal mortality and per capita visits to medical and health institutions. (Since there are no direct data on the number of years of education, referring to the measurement method of the National Bureau of Statistics of China, the average education level here in this study refers to the average number of years of education over 6 years old, and the calculation formula: the average number of years of education over 6 years old = [number of people not attending school × 0 + number of primary school × 6 + number of junior high school × 9 + number of high school students × 12 + (college + undergraduate + graduate) × 16]/population over 6 years old.)

(3) Improvement in the level of health insurance. Health security is the last line of defense for vulnerable populations (the elderly, children, pregnant women and the disabled). Basic health insurance and social health security are the two pillars for improving health security. Basic health insurance is mainly measured by the participation rate of medical insurance, maternity insurance and work-related injury insurance. Social health insurance is reflected through the proportion of government medical and health expenditure, the density of public administration and social security units and per capita social medical assistance expenditure.

(4) Construction level of a healthy environment. A healthy environment is a prerequisite for improving public health. Natural environment construction, living environment construction and public safety construction are its core areas. The construction of the natural environment is mainly measured by the proportion of days with good air quality and forest coverage. The construction of the living environment is mainly reflected in the greening coverage rate of built-up areas and the agricultural non-point source pollution index. The construction of public safety mainly includes two aspects: the number of deaths and injuries per 10,000 people in traffic accidents and the proportion of public safety expenditure.

(5) Development level of the health industry. Developing the health industry is fundamental to ensuring the supply of medical and health materials. The development of the medical and pharmaceutical industry and the recuperation and leisure industry can greatly enhance people’s sense of security and happiness. Among them, the development of the medical and pharmaceutical industry is mainly reflected by the coverage rate of medical and health institutions of 10,000 people, the proportion of the income of medical and health institutions to GDP, the proportion of the main income of pharmaceutical manufacturing industry to GDP, and the per capita ownership of finished pharmaceutical products; the development of the health care and leisure industry is measured by the number of pension beds per 10,000 elderly population and the number of cultural and sports leisure industries per 10,000 people.

### 3.2. Research Methods

#### 3.2.1. Global Entropy Method

The entropy method is a comprehensive evaluation method for objectively weighing multiple evaluation indexes. However, the traditional entropy method can only process cross-sectional data or time series data; in order to further ensure the scientific accuracy of the measurement results, this paper uses Stata 17.0 software and adopts the global entropy method that can handle multiple indicators, multiple years and multiple provinces, and the specific steps are as follows:

① Build a global evaluation matrix. Evaluation of the urban resilience level of β indicators in the M year of α province. The global matrix of αM×β is obtained by arranging M section data sheets $x^{M}={\left(x_{ij}\right)}_{\alpha \times \beta }$ in chronological order, which is recorded as
(1)$$x=\left(x^{1},x^{2},x^{3},\ldots ,x^{M}\right)={\left(x_{ij}\right)}_{\alpha M\times \beta }$$

② Standardization of indicators. The basic indicators in the global matrix have different units, which cannot be calculated directly, and the indicators need to be standardized. Let $x_{ij}$ be the index value of item $j\left(j=1,2,3,\ldots ,n\right)$ in the $i\left(i=1,2,3,\ldots ,m\right)$ evaluation province, specifically:(2)$$\mathrm{Positive}\;\mathrm{indicators}:\;Z_{ij}=\frac{x_{ij}-\min x_{j}}{\max x_{j}-\min x_{j}}$$
(3)$$\mathrm{Negative}\;\mathrm{indicators}:Z_{ij}=\frac{\max x_{j}-x_{ij}}{\max x_{j}-\min x_{j}}$$

In the above formula, $Z_{ij}$ is the standardized index value, $x_{ij}$ is the original value of index j of the i province and $\max x_{j}$ and $\min x_{j}$ are the maximum and minimum values of index j, respectively.

③ Calculate the proportion of indicators. Calculate the proportion $y_{ij}$ of the i province in the index under index j:(4)$$y_{ij}=\frac{Z_{ij}}{\sum_{i=1}^{\alpha M}Z_{ij}},1\leq i\leq \alpha M,1\leq j\leq \beta$$

④ Calculate the information entropy value. Calculate the information entropy value $e_{j}$ of the j index:(5)$$e_{j}=-k\sum_{i=1}^{\alpha M}y_{ij}\ln y_{ij},1\leq i\leq \alpha M,1\leq j\leq \beta ,\;k=\frac{1}{\ln\alpha M}$$

⑤ Calculate the value of information utility. Calculate the coefficient of variation $d_{j}$ of the *j* index: (6)$$d_{j}=1-e_{j}$$

⑥ Calculate index weight $W_{j}$: (7)$$W_{j}=\frac{d_{j}}{\sum_{i=1}^{m}d_{j}}$$

⑦ Calculating the public health level U of China: (8)$$U=\sum_{i=1}^{m}y_{ij}W_{j}$$

#### 3.2.2. Dagum Gini Coefficient

In 1997, Dagum improved the strict constraint conditions (no cross between grouped samples) when measuring the inequality degree by using the Theil index and classical Gini coefficient and proposed the Dagum Gini coefficient and its decomposition model that can further decompose the overall difference into three parts: intra-regional difference, inter-regional net difference and inter-regional hypervariable density, which has become the mainstream method for measuring regional differences [41]. This paper uses MATLAB 2018a software for calculation; the formula is as follows:(9)$$G=\frac{\sum_{j=1}^{k}\sum_{h=1}^{k}\sum_{i=1}^{n_{j}}\sum_{r=1}^{n_{h}}\left|y_{ji}-y_{hr}\right|}{2n_{j}^{2}\bar{y}}$$

In Formula (9), G is the overall Gini coefficient, representing the difference in public health level in China, $y_{ji}$ and $y_{hr}$ represent the public health level of the i province in region j and the r province in region h, $\bar{y}$ represents the average value of regional public health level, n and k represent the number of provinces and regions, $n_{j}$ and $n_{h}$ represent the number of provinces in region j and region h, respectively. In this paper, n is 31, k is 4 (northeast, east, central, west), Gini coefficient $G_{jj}$ of region j and Gini coefficient $G_{jh}$ between region j and region h can be expressed as
(10)$$G_{jj}=\frac{\sum_{i=1}^{n_{j}}\sum_{r=1}^{n_{h}}\left|y_{ji}-y_{hr}\right|}{2n_{j}^{2}\bar{y_{j}}}$$
(11)$$G_{jh}=\frac{\sum_{i=1}^{n_{j}}\sum_{r=1}^{n_{h}}\left|y_{ji}-y_{hr}\right|}{n_{j}n_{h}\left(\bar{y_{j}}+\bar{y_{h}}\right)}$$

In Formulas (10) and (11), $\bar{y_{j}}$ and $\bar{y_{h}}$ are the average public health level for regions j and h. According to the research of Dagum (1997), the overall Gini coefficient G can be further decomposed into three parts: the intra-regional difference contribution $G_{w}$, the inter-regional net difference contribution $G_{nb}$ and the inter-regional hypervariable density contribution $G_{t}$, and $G=G_{w}+G_{nb}+G_{t}$.

#### 3.2.3. Kernel Density Estimation

Kernel density estimation is a nonparametric estimation method that describes the distribution of variables with continuous density curves. It mainly reflects the level, concentration, polarization and difference of urban resilience through the distribution position, peak height and width, peak number and distribution extensibility of the curve. This estimation method is widely used in spatial disequilibrium analysis because of its strong robustness and no assumptions on data [42]. The application of Kernel density estimation to explore the horizontal distribution characteristics of the public health level and regional absolute differences can supplement and improve the relative differences of Dagum Gini coefficient measurement. This paper uses MATLAB 2018a software for calculation. Assume that $f\left(x\right)$ is the density function of the public health level x in China, and the formula is as follows:(12)$$f\left(x\right)=\frac{1}{Nh}\sum_{i=1}^{N}K\left(\frac{X_{i}-\bar{x}}{h}\right)$$

In Formula (12), N is the number of regional provinces, $K\left(x\right)$ is the kernel function, $X_{i}$ is the independent distribution observation value, $\bar{x}$ is the mean value and h is broadband. The narrower the bandwidth, the higher the accuracy. 

#### 3.2.4. Global Spatial Autocorrelation

Global spatial autocorrelation is used to test whether there is a significant correlation or a spatial distribution pattern between the attribute values of a phenomenon and its adjacent units in the geographical space. It can be achieved by calculating the Moran index [43]. This paper uses GeoDa software for calculation, and the formula is as follows:(13)$$I=\frac{\sum_{i=1}^{n}\sum_{j\neq i}^{n}W_{ij}\left(x_{i}-\bar{x}\right)}{S^{2}\sum_{i=1}^{n}\sum_{j=1}^{n}W_{ij}x_{i}x_{j}/\sum_{i=1}^{n}\sum_{j=1}^{n}x_{i}x_{j}}$$ where n is the number of spatial units, $x_{i}$ and $x_{j}$ represent the values of spatial element x in spatial units i and j, $\bar{x}$ is the mean of element x and $W_{ij}$ is the spatial weight matrix (queen adjacency spatial weight matrix is used in this paper). Moran’s *I*∈[−1,1], when Moran’s *I* > 0 means that the regions are positively correlated, when Moran’s *I* = 0 or close to 0 means that the regions are spatially uncorrelated and when Moran’s *I* < 0 means that the regions are negatively correlated.

#### 3.2.5. Convergence Model

In order to further investigate the evolution characteristics of regional differences in public health level in China, this paper uses MATLAB 2018a software and adopts two types of methods: σ convergence and β convergence, which are tested from the perspectives of stock and increment.

σ convergence refers to the trend that the deviation of the resilience level of each province decreases with time. In this paper, the variation coefficient is used to describe the status of σ convergence. The formula is as follows:(14)$$\sigma =\frac{\sqrt{\sum_{i=1}^{n_{j}}\left(PH_{ij}-\bar{PH_{ij}}\right)/n_{j}}}{\bar{PH_{ij}}}$$

In Formula (14), $PH_{ij}$ (Public Health) represents the public health level of i province in region j, $\bar{PH_{ij}}$ represents the average public health level of j provinces and $n_{j}$ represents the number of provinces in region j.

β convergence means that over time, the growth rate of provinces with low public health levels will gradually catch up with those with high public health levels, and the gap between the two will narrow and become consistent. β convergence can be divided into absolute β convergence and conditional β convergence. Absolute β convergence means that there is a convergence trend without considering a series of economic and social factors such as Economic development level, financial pressure, urbanization level, population density and opening up level that have an important impact on public health level, while condition β convergence considers or controls a series of important economic and social factors. Considering that there may be spatial spillover effects on the public health level and bias in traditional OLS estimates, this study adopts a β-converged spatial econometric model; commonly used spatial econometric models mainly include spatial lag model (SLM), spatial error model (SEM) and spatial Dubin model (SDM), according to the spatial econometric model selection steps proposed by Elhorst (2014): first of all, according to the LM test, determine whether there is a spatial effect on public health level in the country and four major regions (determine whether to choose OLS or spatial econometric model). Secondly, the specific forms of the spatial model (SLM, SEM and SDM) are determined according to the LR test and Wald test. Finally, the specific random effect or fixed effect (random effect, space fixed, time fixed and two-way fixed effect) is selected according to the Hausman test results [44]. The absolute β convergence formula is as follows:(15)$$OLS:\ln\left(\frac{PH_{i,t+1}}{PH_{i,t}}\right)=\alpha +\beta \ln PH_{it}+\mu_{i}+\eta_{t}+\varepsilon_{it}$$
(16)$$SAR:\ln\left(\frac{PH_{i,t+1}}{PH_{i,t}}\right)=\alpha +\beta \ln PH_{it}+\rho \sum_{j=1}^{n}w_{ij}\ln\left(\frac{PH_{i,t+1}}{PH_{it}}\right)+\mu_{i}+\eta_{t}+\varepsilon_{it}$$
(17)$$SEM:\ln\left(\frac{PH_{i,t+1}}{PH_{i,t}}\right)=\alpha +\beta \ln PH_{it}+\mu_{i}+\eta_{t}+u_{it}\;u_{it}=\lambda \sum_{j=1}^{n}w_{ij}\ln\left(\frac{PH_{i,t+1}}{PH_{it}}\right)$$
(18)$$SDM:\ln\left(\frac{PH_{i,t+1}}{PH_{i,t}}\right)=\alpha +\beta \ln PH_{it}+\rho \sum_{j=1}^{n}w_{ij}\ln\left(\frac{PH_{i,t+1}}{PH_{it}}\right)+\gamma \sum_{j=1}^{n}w_{ij}\ln\left(PH_{it}\right)+\mu_{i}+\eta_{t}+\varepsilon_{it}$$

In Formulas (15)~(18), $PH_{i,t+1}$ represents the urban resilience of the i province in the t+1 period, $UR_{i,t}$ represents the public health level of the i province in the t period and $\ln\left(\frac{PH_{i,t+1}}{PH_{i,t}}\right)$ represents the growth rate of the i public health level in the t+1 period. β is the convergence coefficient, β<0 indicates that the regional province toughness has a convergence trend, and vice versa; there is a divergence trend, and the convergence rate is $-\ln\left(1-\left|\beta \right|\right)/T$. ρ is the spatial lag coefficient, λ is the spatial error coefficient, γ is the spatial autocorrelation coefficient of the independent variable, w, $\mu_{i}$, $\eta_{t}$ and $\varepsilon_{it}$, respectively, represent the spatial weight matrix, regional effect, time effect and random disturbance term. Condition β convergence is to add a series of control variables on the basis of absolute β convergence.

Referring to existing research results, seven control variables are selected: economic development level, financial pressure, urbanization level, population density, advanced industrial structure level, scientific and technological innovation level and opening-up level. Specifically, the level of regional economic development is expressed in terms of GDP per capita. The ratio of local fiscal public budget expenditure to public budget revenue is used to express the situation of regional financial pressure. The proportion of the urban population to the permanent population at the end of the year is used to represent the level of regional urbanization. The population density of the region is characterized by the number of people per unit area. The proportion of tertiary industry output value to GDP is used to determine the level of regional advanced industrial structure. The number of patents granted by 10,000 people indicates the level of regional scientific and technological innovation. The proportion of foreign investment in GDP actually utilized represents the level of regional opening up.

### 3.3. Data Collection and Pretreatment

The sample of this study is the data of 31 provinces (municipalities directly under the central government and autonomous regions) in China since the new medical reform (2009–2020) (Hong Kong, Macao and Taiwan are not included in the assessment sample due to lack of data), and it is divided into four regions: eastern, northeastern, central and western. The index data and variable data involved in this study are mainly derived from the China Health Statistical Yearbook, China Statistical Yearbook, China Industrial Statistical Yearbook, China Basic Unit Statistical Yearbook, China Education Statistics Yearbook and the statistical yearbooks and statistical bulletins of various provinces from 2010 to 2021, among which the missing data of individual years and individual provinces were completed by linear function method (TREND function). In addition, the currency numerical index data involved in this paper are based on 1978, and they are treated with constant prices using corresponding price indexes.

## 4. Research Results

### 4.1. Spatial and Temporal Distribution of Public Health Level in China

#### 4.1.1. Time Evolution

The global entropy method is used to calculate the provincial public health level and its dimensional status in China from 2009 to 2020, and the specific results are shown in Figure 1. As can be seen from Figure 1, China’s provincial public health level presents the following characteristics:

(1) The overall level of public health in China is relatively low, but there is a steady upward trend. From 2009 to 2020, China’s provincial public health level was between 0.1699 and 0.3183, which was generally a low level overall, but it can be found that China’s provincial public health level increased by 0.1484 in 12 years, with an average annual growth rate of 7.28%, showing an overall good development trend. Especially after 2010, the accelerated growth of the public health level also proves that the “new medical reform” has good policy effect, promotes the development of China’s medical and health undertakings, improves the people’s public health level and is conducive to the realization of the strategic goal of a “healthy China”.

(2) The development of the dimensions of China’s public health level is uneven. ① The development of the health industry and the optimization of health services are the core driving forces for promoting the improvement of China’s public health level since the new medical reform. Specifically, the development level of the health industry increased from 0.0266 in 2009 to 0.0788 in 2020, an increase of nearly three times in 12 years, with an average annual growth rate of 24.698%, and after 2015, it surpassed the level of health security perfection and was in a leading position in various sub-dimensions. Although the level of health service optimization started from the lowest starting point, only 0.0207 in 2009, the growth rate during the new medical reform period was relatively fast, and after 2016, it even surpassed the sub-dimension of healthy environment construction and the sub-dimension of healthy life popularization, and in 2019 it was basically close to the sub-dimension of health security improvement, and the overall level was 0.0642, and there was a slight decline in 2020 due to the impact of the COVID-19 epidemic. ② The improvement of health security and the popularization of healthy life is the backbone of China’s public health improvement. Specifically, the improvement level of health security has been showing a steady upward trend since the beginning of the “new medical reform”, especially after the 18th generation of the Communist Party of China. This dimension has been leading other sub-dimensions since it was not implemented in 2010–2015, and the contribution rate to China’s public health level in 2020 is still as high as 22.87%. The popularization of healthy living basically maintained a steady upward trend during the “new medical reform” period, from 0.0317 in 2009 to 0.0553 in 2020. ③ The construction level of a healthy environment has become a shortcoming in the improvement of China’s public health level since the new medical reform. The starting point of the level of healthy environment construction is the highest, 0.0471 in 2009, ahead of other sub-dimensions, but it was quickly surpassed, and after 2017, it began to rank last, and the overall level in 2020 was only 0.0495, with an average annual growth rate of less than 0.02%, as a precursor to the improvement of China’s public health level, it needs to be focused on in the future.

#### 4.1.2. Spatial Distribution

In order to explore the spatial distribution of public health in China since the new medical reform, on the one hand, according to the national administrative regions, 31 provinces were divided into four major regions: northeast, east, central and western, and the level of public health in each region and its sub-dimensional status were explored (Figure 2 and Figure 3). On the other hand, in order to visually display a more detailed spatial and temporal distribution pattern of China’s public health level since the new medical reform, combined with the natural breakpoint method and the principle of an equal interval that comes with ArcGIS, the public health level is divided into four levels: lowest level (less than or equal to 0.1500), low level (less than or equal to 0.3000 and greater than 0.1500), medium level (less than or equal to 0.4500 and greater than 0.3000) and high level (greater than 0.4500), and select 2009, 2012 (the 18th National People’s Congress of the Communist Party of China), 2017 (the 19th National People’s Congress of the Communist Party of China) and 2020; ArcGIS 10.8 software was used to visualize the public health level of 31 provinces, as shown in Figure 4.

(1) From the overall level of public health, public health in the four major regions of China showed an upward trend, but there were significant regional differences. From 2009 to 2020, the public health level in the four major regions showed a steady growth trend, with the average annual growth rates of 0.89%, 1.44%, 1.12% and 1.21% in the northeast, east, central and western regions, respectively, which means that the overall development trend of China’s public health level is improving. From the perspective of regional differences, the public health level of the four major regions showed a distribution pattern of “eastern > northeastern > central > west” before 2015 and showed “eastern > northeast > western > central” after 2015, but it is worth noting that in 2020, the northeast region may have a downward trend due to the superimposed impact of economic development and the COVID-19 epidemic, and the growth rate of the western region also began to slow down. In general, the difference in public health levels between the four major regions has gradually expanded, and the eastern region has a relatively high level of public health and a faster growth rate due to its superior geographical location and strong economic development strength. However, the public health level in the northeast, central and western regions has grown slowly, and the northeast region has even declined, and it is necessary to increase the investment of medical and health resources in the northeast, central and western regions in the future and gradually narrow the gap between the northeastern, central and western regions and the eastern region.

(2) From the perspective of the level of public health in different dimensions, the level of public health in the four major regions is significantly different. Specifically, the five dimensions in the eastern region are significantly different, but the overall is high, and the dimensions basically show the characteristics of “the improvement level of health insurance > the development level of the health industry > the optimization level of health services > the construction level of healthy environment > the popularization level of healthy life”, that is, the optimization of health services, the construction of healthy environment and the popularization of healthy life are the shortcomings of the eastern region. The differences in the five dimensions of the northeastern, central and western regions were small, but all were at a low level (below 0.08). Specifically, the northeast region has the characteristics of “the development level of health industry > the improvement level of health insurance > the construction level of healthy environment > the optimization level of health services > the popularization level of healthy life”, which also requires that the northeast region should comprehensively improve the level of all dimensions, focus on the popularization of healthy life and the construction of healthy environment, and increase the publicity of healthy lifestyle and ecological, environmental protection. The characteristics of the central region are basically consistent with those of the eastern region, and it is also necessary to focus on monitoring the popularization level of healthy living and the construction of a healthy environment; “The development level of the health industry > the popularization level of healthy living > the construction level of the health environment > the optimization level of health services > the improvement level of health insurance” is presented within each dimension of the western region. The optimization of health services and the improvement of health security are the shackles that need to be broken through in the western region.

(3) From a specific province perspective, since the new medical reform, the public health level in China has undergone a transformation from “lowest-level and low-level contiguous, medium-level sporadic and high-level none” to “lowest level disappearing, low level and medium level contiguous, and high level sporadic”, showing a good overall development trend. Specifically, ① in 2009, China’s provincial public health level presented a distribution pattern of “lowest level and low level contiguous, medium level sporadic, and high level absent”. The only medium-level provinces are Shanghai and Beijing in the east. The number of low-level provinces is 16, accounting for 51.61% of the total, and the remaining 13 provinces are lowest-level, mainly concentrated in the central and western regions. ② In 2012, China’s public health level showed a distribution pattern of “low-level contiguous, high-level, medium-level and lowest-level sporadic distribution”. The high level is only Beijing; the medium level includes Shanghai and Zhejiang provinces; the lowest level is only the western Tibet and Guizhou provinces; the remaining 26 provinces are all low-level states, of which 14 provinces have been transformed from lowest-level states, and the level of public health has improved as a whole. ③ In 2017, China’s public health level showed a “pyramid” of “lowest-level disappearing, low level contiguous, medium and high level sporadic”. Shanghai has risen to a high-level echelon. the medium-level has also increased from the original two provinces to five provinces, the additional provinces are Tianjin, Jiangsu, Shandong and Guangdong, the remaining 24 provinces are all low level, all lowest level provinces are listed, and the level of public health continues to improve. ④ In 2020, China’s public health level showed a distribution pattern of “low-level and medium-level contiguous, and high-level sporadic”. The high level is still only two provinces, Beijing and Shanghai. The number of medium-level provinces has increased to 10, and the remaining 19 provinces are all low levels, and China’s public health level has improved by leaps and bounds.

Note: This map is based on the GS (2020)4630 standard map downloaded from the standard map service system of the Ministry of Natural Resources of China, and the base map has not been modified.

### 4.2. Regional Differences and Decomposition of Public Health Levels in China

In order to further explore the trend of regional differences in China’s public health level and its main sources, the Dagum Gini coefficient and its decomposition model were used to measure its regional differences (intra-regional differences and inter-regional differences) and main sources of differences (intra-regional difference contribution, inter-regional difference contribution and hypervariable density contribution), and the specific results are shown in Figure 5 and Figure 6 and Table 2.

(1) Intra-regional differences in China’s public health level. From a national perspective, the overall Gini coefficient of public health levels from 2000 to 2020 showed a trend of decreasing first and then increasing “V-shaped”, decreasing from 0.1633 in 2000 to 0.1107 in 2015 and then showing a slight upward trend, rising to 0.1230 in 2017 and basically maintaining a level of around 0.12 after 2017. In 2019, there was even a slight increase, but in 2020, it continued to decline, and the overall difference showed a gradually narrowing trend. However, there is a need to prevent recurrence from continuing. From the perspective of the four major regions, the regional Gini coefficient of the public health level in the eastern region exceeds the overall level of the country, which also shows that the public health level in the eastern region varies greatly, showing a good development trend of “V-shaped” that first decreases and then rises. It first decreased from 0.1807 in 2009 to 0.1367 in 2015 and then continued to rise to 0.1619 in 2020, so it is necessary to focus on the public health level of the provinces in the eastern region in the future to prevent polarization. The regional difference in public health level between the central region and the western region was generally small and showed a fluctuating downward trend, from 0.0692 in 2009 to 0.0336 in 2020 in the central region and 0.0740 in the western region from 0.0740 in 2009 to 0.0552 in 2020, and the difference in public health level between provinces in the region is decreasing. The northeast region has the smallest regional differences in public health levels, but it is worth noting that there is a fluctuating upward trend, from 0.0262 in 2009 to 0.0295 in 2020, which needs to be watched in the future.

(2) Inter-regional differences in China’s public health level. The inter-regional Gini coefficient ranking of the public health level of the four major regions in China is as follows: eastern–western (0.1949) > eastern–central (0.1836) > northeastern–eastern (0.1342) > northeastern–western (0.0953) > northeastern–central (0.0800) > central–western (0.0611), except for the northeast–east, the overall fluctuation between the other two regions showed a decline in the overall fluctuation, but it should be noted that the Gini coefficient of the eastern–western and eastern–central public health levels showed a steady expansion trend after 2015. The “V-shaped” development trend of the Gini coefficient between the northeastern and the eastern and the public health level first decreases and then rises, and the Gini coefficient between the two regions in 2020 is 0.1605 larger than the 0.1455 in 2009, which may be due to the impact of climate and economic and social development, the overall public health level of the northeast region declines, while the eastern region still maintains a good development trend, so the difference between the two regions will continue to expand. The regional differences in public health levels in the northeastern–central and central–western regions were small and showed a steady downward trend. Eastern–central decreased by 0.0851 from 0.1231 in 2009 to 0.0380 in 2020. Central–western decreased by 0.0287 from 0.0761 in 2009 to 0.0474 in 2020.

(3) Regional differences in China’s public health level and its decomposition. Inter-regional differences are the main source of regional differences in China’s public health level, but their contribution rate shows a downward trend, while the contribution rate of intra-regional differences shows an upward trend, and the contribution rate of hypervariable density is basically at a low level below 10%. Specifically, from the perspective of intra-regional net differences, the intra-regional net differences in China’s public health level have shown a fluctuating downward trend over the past 12 years, from 0.0359 in 2009 to 0.0296 in 2020. At the same time, as the overall regional differences across the country are shrinking, their contribution to the regional differences in China’s public health level is increasing, from 21.97% in 2009 to 24.40% in 2020. From the perspective of inter-regional net difference, the inter-regional net difference in China’s public health level has shown a fluctuating downward trend over the past 12 years, decreasing from 0.1155 in 2009 to 0.0800 in 2020 and a decrease of 0.0355. Its contribution rate to the regional difference in China’s public health level has also decreased from 70.76% in 2009 to 66.09% in 2020, but it has always been the main source of regional differences in China’s public health. From the perspective of hypervariable density, the overall hypervariable density of China’s public health level during the past 12 years is relatively low. The hypervariable density mainly reflects the overall contribution of overlapping parts between regions. The low hypervariable density also indicates that the four regional division methods are effective ways to divide provinces and achieve reasonable clustering. In addition, it can be found that the horizontal hypervariable density of public health water is in a slow downward trend, and its contribution rate to regional differences in China’s public health level remains basically below 10%. To sum up, in the future, it is necessary to focus on regional differences in public health levels while taking effective measures to prevent the expansion of regional differences.

### 4.3. Dynamic Evolution of Public Health Levels in China

Dagum Gini coefficient and its decomposition reveal the overall difference in public health level in China and its main sources and reveal the relative difference trend between the four major regions. In this part, the Kernel density estimation method will be used to reveal the dynamic time evolution characteristics of China’s public health level and its sub-dimensions with the help of the distribution position of the Kernel density curve, the distribution status of the main peak, the ductility of the distribution and the number of peaks, etc., as shown in Table 3 and Figure 7.

(1) Distribution location. In 2009, when a new round of medical reform began, China continued to increase its investment in resources in the field of health care. However, since the overall improvement of environmental construction is a project with large investment, little return and long time consumption, the Kernel density curve of the level of health environment construction remains basically unchanged. In addition to the impact of the COVID-19 epidemic in 2020, China’s public health level and the Kernel density curve of other dimensions show a trend of moving to the right; however, the movement is relatively small, which also indicates that the public health level in most provinces of China is on an upward trajectory, and the construction of a “healthy China” is steadily advancing. In addition, it is also necessary to pay special attention to the recent left shift in the popularity of healthy living and the overall public health level. In the future, it is necessary to provide guidance, increase the promotion of healthy lifestyles, promote national fitness activities and help people establish a healthy life concept.

(2) Pattern of main peak distribution. The main peak height of the overall public health level and the development level of the health industry and the Kernel density curve decreases and the width becomes larger, which means that the dispersion degree of the public health level and the development level of the health industry in the entire sample period is on the rise, which is caused by the increase in the allocation of medical and health resources in different regions and the different degrees of development of the medical and health industry between regions, so it is necessary to increase public health investment in low-level areas and help the development of the health industry in the future. The height of the main peak of the Kernel density curve of the health security improvement level and the health environment construction level increased and the width became smaller, which meant that the regional difference between the health security improvement level and the healthy environment construction level was narrowing during the entire sample period, which was closely related to the promotion of new medical reform policies and the strengthening of ecological protection. The main peak height of the Kernel density curve of the popularization level of healthy life and the optimization level of health service “first decreases and then rises”, and the width is “first wide and then narrow”, which means that the absolute difference between the popularization level of healthy life and the optimization level of health services expands first and then narrows during the entire sample period, especially after 2015. This empirical result proves that in recent years, under the guidance of the national strategy, provinces have continuously strengthened health service publicity and medical and health service improvement, and the gap between regions is narrowing. 

(3) Distribution ductility. The Kernel density curves of the public health level and dimension showed significant right-tailing characteristics; that is, the public health level and dimension level of some provinces in the region were significantly higher than other provinces in the same region. For example, the public health and dimension scores of Beijing, Shanghai, Zhejiang and other provinces in the east are relatively high, which makes the public health level and dimension distribution curve show the characteristics of right tailing. At the same time, further analysis shows that the right tailing fluctuates shortened; that is, there is a convergence trend in the ductility of the distribution, which means that the probability of extreme values in the public health level and various dimensions is low.

(4) Number of peaks. The number of peaks in China’s public health level and its dimensional Kernel density curve is always one, which shows that there is no polarization phenomenon in both the overall state of public health and its dimensional level. Further monitoring and restraint are needed to prevent polarization or multi-polarization and reduce public health inequalities.

### 4.4. Spatial Convergence of the Public Health Level in China

#### 4.4.1. σ Convergence

As Figure 8 shows that the public health level of the national, eastern, central and western regions showed a σ convergence trend, and the national and eastern regions had a rebound trend after 2015, while the northeast region showed a divergent trend. From a national perspective, the overall coefficient of variation of public health levels in 2009–2020 showed a “V-shaped” trend of first decreasing and then increasing. Specifically, from 2009 to 2015, the national public health level showed a fluctuating downward trend, decreasing from 0.3551 in 2009 to 0.2480 in 2015, a decrease of 0.1071. From 2015 to 2020, the overall steady increase reached a peak of 0.2905 in 2020, which may be due to the proposed “Healthy China 2030” plan and increased investment in medical and health resources in various regions. Some provinces have developed rapidly, and the gap between regions has further widened. From the eastern, the coefficient of variation of regional public health level during the sample period is higher than that of the whole country and the other three major regions, and its overall trend of change is consistent with that of the national, showing a “V-shaped” trend of first falling and then rising, indicating that the differences in public health levels among provinces in the region are gradually expanding after 2015. From the perspective of the central and western regions, the overall coefficient of variation of regional public health level during the sample period is relatively low and shows a fluctuating downward trend. The public health level in the central and western regions presents a σ convergence trend. From the perspective of northeast China, the regional public health level showed a fluctuating upward trend during the sample period, rising from 0.0591 in 2009 to 0.0678 in 2020. There was a slight divergence trend within the region, which requires special attention in the future to prevent the intensification of regional polarization.

#### 4.4.2. Spatial Correlation

Before judging whether China’s public health level has the characteristics of β convergence, it is necessary to borrow the global autocorrelation method to explore its spatial correlation relationship. Therefore, this study uses the Moran index to conduct a preliminary test of the correlation of public health levels in 31 provinces in China, and the results are shown in Table 4. From 2009 to 2020, the Moran index of China’s public health level was significantly positive, indicating that there was a significant positive correlation between China’s public health level, that is, China’s public health level did not appear randomly in space, but showed the characteristics of “high-high” or “low-low” spatial agglomeration. At the same time, a significantly positive Moran index also indicates that the level of public health in a province depends not only on its own factors but also on its surroundings. From 2009 to 2020, the Moran index of China’s public health level showed a fluctuating downward trend, from 0.296 in 2009 to 0.211 in 2020, but the Moran index was significantly greater than 0 throughout the sample period, indicating that geographical location has become one of the important factors affecting China’s public health level.

#### 4.4.3. β Convergence

(1) Absolute convergence. Table 5 shows the absolute β convergence test results and the corresponding convergence speed of the national and four major regional public health levels. First, according to the LM test, it is judged whether the absolute convergence of the national and regional public health levels has a spatial effect. Secondly, the specific form of the spatial model (whether the spatial Dubin model will degenerate into a spatial lag model and a spatial error model) is determined according to the LR test and Wald’s test. Again, random-effect or fixed-effect results were selected based on the results of the Hausman test. Finally, the specific forms of fixed effects (spatial fixation, temporal fixation and bidirectional fixation effects) are selected according to whether the spatial fixation effect and the temporal fixation effect pass the test. The specific results are as follows: first, there is absolute β convergence in the public health level of the whole country and the four major regions, except for the insignificant convergence coefficient in the northeast region, the convergence coefficient in the whole country and other regions is significantly negative at the confidence level of 1%, that is, without considering the influence of economic, social and natural factors, the regional differences in the public health level of the national, eastern, central and western regions will gradually narrow, which is also consistent with the trend of gradual narrowing of its coefficient of variation. Second, there are differences in the absolute β convergence speed of the national and four major regional public health levels; the convergence speed of the whole country is 3.7815%, only the convergence speed in the northeast region is lower than the national convergence speed, only 2.4519%, and the other three regions are higher than the national convergence speed, of which the central region is the highest, reaching 8.2372%. Third, the whole country and the northeast region have different spatial effects. Specifically, the coefficient of public health level in the whole country and the northeast region has passed the significance level test of 10%, indicating that the rate of change of public health level in the whole country and northeast region will be affected by the change rate of public health level and public health level in other regions, but the national coefficient is significantly positive, which indicates that the national public health level change rate will increase with the increase of the change rate in other regions, and the coefficient in the northeast region is significantly negative, which is the opposite of the national situation. However, the above characteristics of absolute convergence of public health levels in the whole country and the four major regions are established under the strong assumption that the economic and social conditions of each province are similar, but the reality is not the same, and the resource endowment and economic and social development of different regions are quite different, so it is necessary to further control such factors and carry out conditions convergence for further exploration.

(2) Condition β converges. Table 6 shows the results of the condition β convergence test for the national and four major regional public health levels, and the model selection process is consistent with absolute β convergence, which will not be repeated here. The specific results showed that, first, there was condition β convergence at the national and regional public health levels, and the convergence coefficient was significantly negative at the confidence level of 1%. This shows that after considering a series of economic and social factors such as economic development level, financial pressure, urbanization level and population density, the public health level of northeast China and the national, eastern, central and western regions is consistent, and all show a convergence trend. Second, except for the central region, the convergence rate of condition β of the national and other three major regions of public health is higher than that of absolute β, among which the convergence rate of the national, northeastern, eastern and western regions increased by 2.2900%, 2.4595%, 5.6371% and 3.170%, respectively, which further explains the scientific nature of the selected control variables. Third, the level of public health in the whole country and the four major regions shows different spatial effects. The coefficient of the national public health level is still significantly positive, suggesting that the improvement of the public health level in some provinces nationwide will promote an increase in convergence speed. The coefficient of public health level in the northeast, eastern, central and western regions was significantly negative at the level of 1%, indicating that within the four major regions, the improvement of public health level in some provinces will reduce the convergence rate of the entire region, leading to further widening of the overall difference within the region. 

In addition, it should be noted that there are significant differences in the influencing factors of public health levels across the country and four major regions. After adding a series of control variables, such as economic and social variables, to the conditional β convergence analysis, the sum of the Log-likelihood coefficients across the country and four major regions has increased compared to the absolute β convergence, which further proves the scientific nature of the selection of control variables. From a statistical perspective, there are significant differences in the influencing factors of the change rate of public health levels across the country and the four major regions. Taking the whole country as the research object, the financial pressure and population density situation is significantly negative at the level of 5%, indicating that financial pressure and population density expansion will heterogeneity their convergence speed, leading to regional expansion of public health levels. In addition, for the four major regions, the impact of economic development level, financial pressure, urbanization level, population density, advanced industrial structure level, scientific and technological innovation level and opening-up level on the convergence of regional public health level has significant heterogeneity. Taking the level of economic development as an example, its impact on the change rate of public health levels in northeastern, eastern and western regions is significantly positive at a level of 10%, indicating that the improvement of the economic development level will accelerate its convergence rate and narrow the regional differences in public health levels in northeastern, eastern and western regions. The impact of economic development level on the change rate of public health level in the western region is negative. This is mainly because the western region is located in inland China, with a relatively fragile ecological environment, and economic development is mostly at the cost of ecology and resources. However, the deterioration of the environment will have a negative impact on people’s physical health, ultimately having a negative impact on the regional public health level, leading to further expansion of regional differences.

## 5. Conclusions and Policy Suggestions

### 5.1. Conclusions

Based on the overall requirements of China’s public health put forward by the “Healthy China 2030” Planning Outline, this study constructs the evaluation index system of China’s public health level from five dimensions: the popularization level of a healthy life, the optimization level of health services, the improvement level of health insurance, the construction level of a healthy environment and the development level of the health industry and uses the mainstream objective empowerment entropy method to measure the public health level of 31 provinces in China from 2009 to 2020. Secondly, the Dagum Gini coefficient was used to explore the regional differences and main sources of public health in China. Thirdly, the Kernel density function is used to characterize China’s health level and its dynamic evolution by dimension. Finally, the exploratory spatial data analysis method is used to explore the spatial correlation of the public health level in China, and the convergence characteristics of the public health level in China and four major regions are tested by using a coefficient of variation (σ convergence) and spatial econometric model (β convergence). The main conclusions are as follows:

Firstly, the overall level of public health in China is relatively low, and there is a significant imbalance in regional development, presenting a spatial distribution pattern of “high in the east and low in the central and western regions”. Further research has found that the structural issues in China’s public health level are prominent, mainly reflected in the development of the health industry and the optimization of health services, which are the core driving forces for promoting the improvement of China’s public health level since the new medical reform. The improvement of health insurance and the popularization of healthy life are the backbones of China’s public health improvement. The construction of a healthy environment has become a shortcoming that hinders the improvement of China’s public health level since the new medical reform.

Secondly, the overall regional differences in China’s public health level show a “V-shaped” downward trend of first decreasing and then increasing. Among them, the inter-regional differences are the main sources of the overall regional differences in China’s public health level, and the differences between the northeastern and eastern regions are expanding. In addition, the contribution rate of intra-regional differences to the overall regional differences in China’s public health is on the rise, and special attention needs to be paid to prevent the expansion of the overall regional differences in China’s public health level.

Thirdly, except that the construction of a healthy environment remains basically unchanged, China’s public health level and its sub-dimensions are on an upward trajectory (excluded by the impact of the COVID-19 pandemic in 2020), and there is no polarization. However, there are some provinces within the region that have significantly higher levels of public health and their sub-dimensions than other provinces in the unified region, and the overall level of public health in China and the degree of dispersion of the sub-dimensions of health industry development are on the rise.

Fourthly, from the perspective of σ convergence, except for the northeast region, the public health levels of the whole country and the other three major regions will show a certain convergence trend over time. From the perspective of β convergence, without considering economic and social factors, its convergence characteristics are basically consistent with convergence, while the convergence characteristics in northeast China are not significant, indicating the possibility of regional differences expanding. After considering economic and social factors, the public health level across the country and the four major regions shows a convergence trend. In addition, the impact of economic development level, financial pressure, urbanization level, population density, advanced industrial structure, scientific and technological innovation level and opening-up level on the convergence of public health level in the four major regions is significantly heterogeneous. 

### 5.2. Policy Suggestions

Over the past decade of the “new medical reform”, driven by a series of strategic implementation and related reforms, China’s public health industry has made certain achievements, with significant development in areas such as healthy living, health services and health insurance. However, according to the previous measurement results, it can be found that China’s public health level is still at a relatively low level, with prominent regional imbalances, and there is a trend of further expansion. Therefore, in the future, it is still necessary to promote the process of China’s public health construction from an all-round deep level and wide range of fields. Based on this, this study proposes the following policy recommendations:

Complement weaknesses in the construction of a healthy environment and address structural conflicts to public health levels. China’s public health level is still at a relatively low level, and the low level of health environment construction is a shackle to the overall improvement of China’s public health. Therefore, complementing the shortcomings of health environment construction has become a key link in the improvement of China’s public health level. Prevention is an upstream link in the medical and health system. Strengthening the construction of a healthy environment is a disease prevention measure that can reduce the probability of infection and transmission of infectious diseases from the source. In the future, China needs to continue to adhere to the green development concept of “green water and green mountains are golden mountains and silver mountains”; regard ecological, environmental protection as the key content of preventive public health work; focus on “eradicating” the “harmful soil” that causes infectious diseases and realize the effective connection between upstream “prevention” and downstream “treatment”.Narrow regional development gaps and work together to improve public health. Increase investment in the public health sector in the northeast, central and western regions, and continuously narrow the development gap with the eastern region, especially with the help of the leading provinces in the region, to achieve the coordinated development goal of “connecting points with lines, and leading areas with lines”. For the eastern regions with relatively high levels of public health, on the one hand, it is necessary to accelerate the construction of public health in low-level provinces in the region, focusing on the coverage of medical and health services and guarantees for vulnerable groups in low-level areas. On the other hand, maintain the development momentum of high-level provinces in the region, fully summarize the existing construction experience, form a development paradigm and form point-to-point assistance with provinces with low public health levels in the northeast, central and western regions to help them optimize the allocation of medical resources and rational layout of the medical industry. In short, it is necessary to pay attention to both provinces with low public health and high public health provinces, continuously narrow the development gap between regions and achieve the improvement of public health level in the whole region of China.

In addition, this study has certain limitations. On the one hand, due to the vast area of most provinces, there are significant differences in the basic conditions for the development of internal public health levels. Therefore, there are still certain shortcomings in examining the regional differences, dynamic evolution and convergence of China’s public health level only at the provincial level. In the future, more effective microdata can be obtained through questionnaire surveys and interviews at smaller spatial scales, such as cities, counties and villages, providing data support for more targeted promotion of China’s public health level. On the other hand, this study only attempted to preliminarily explore the basic characteristics of regional differences, dynamic evolution and convergence of provincial public health levels in China. In the future, the mesomeric effect, double difference and other methods can be used to further explore the deep impact mechanism and policy net effect of the improvement of China’s public health level so as to improve the accuracy of the assessment.

## Figures and Tables

**Figure 1 healthcare-11-01459-f001:**
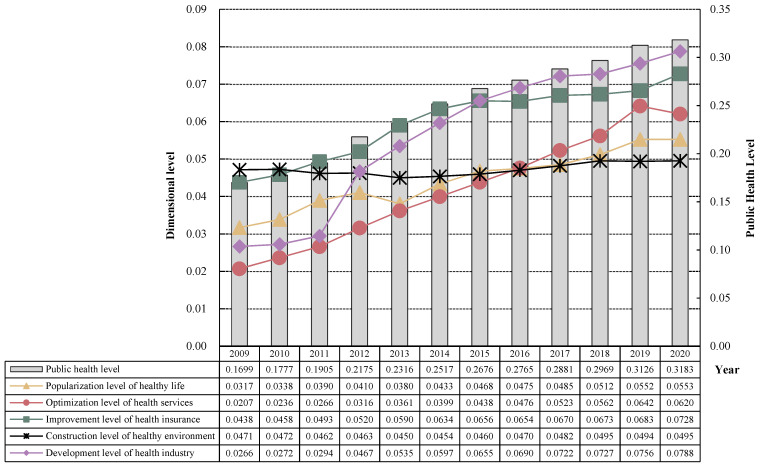
Provincial public health level and its dimensional evolution trend in China from 2009 to 2020.

**Figure 2 healthcare-11-01459-f002:**
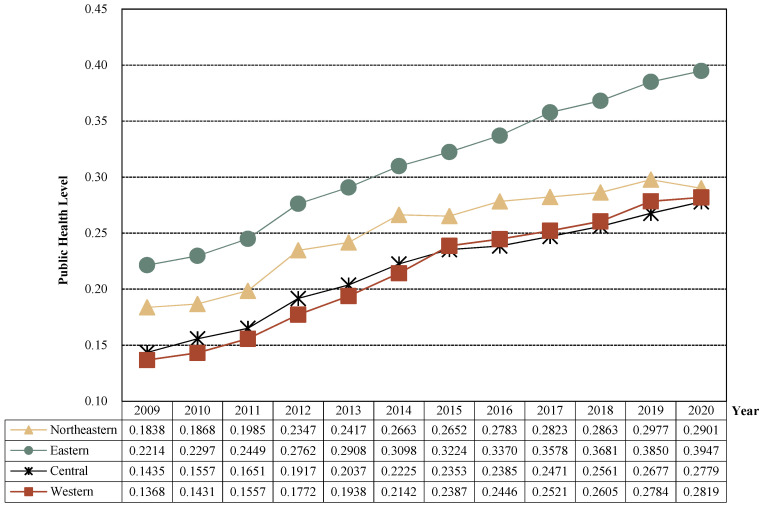
Evolution trend of public health level in four major regions of China from 2009 to 2020.

**Figure 3 healthcare-11-01459-f003:**
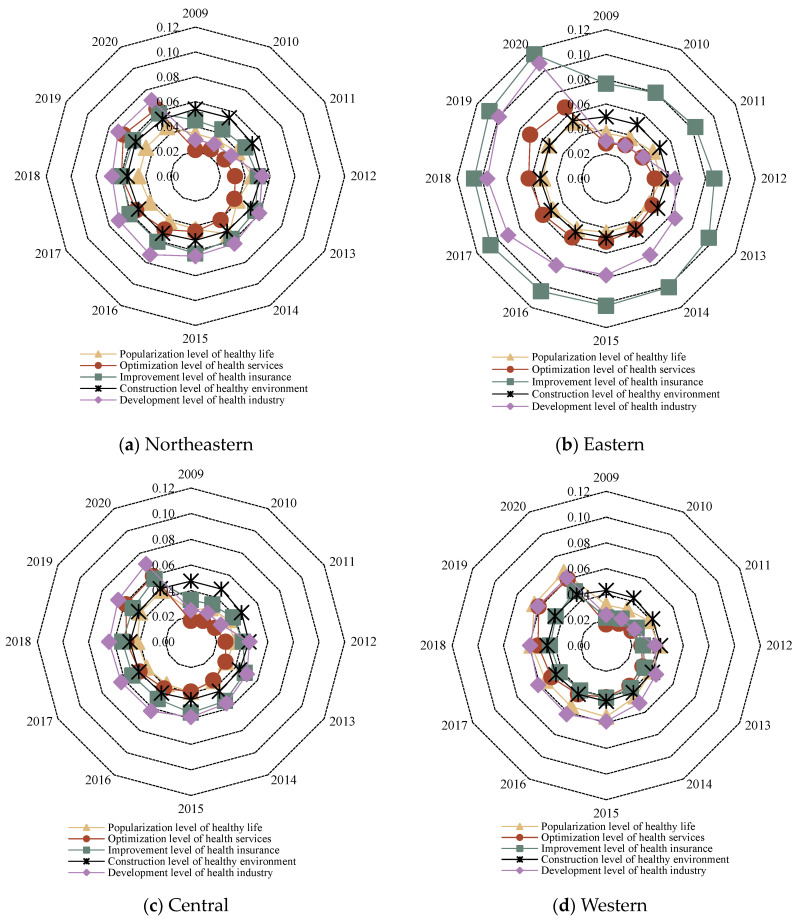
Dimensional changes in public health levels in four major regions of China from 2009 to 2020.

**Figure 4 healthcare-11-01459-f004:**
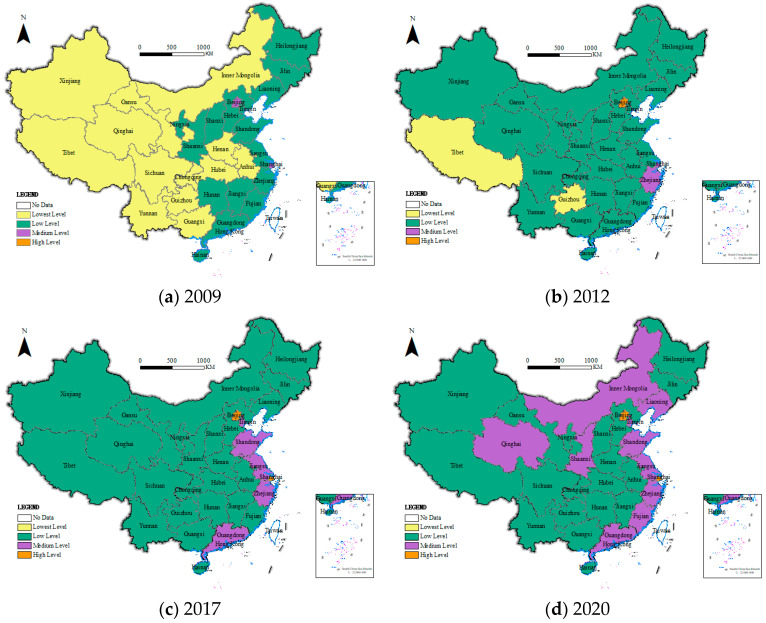
Spatial distribution of provincial public health levels in China from 2009 to 2020.

**Figure 5 healthcare-11-01459-f005:**
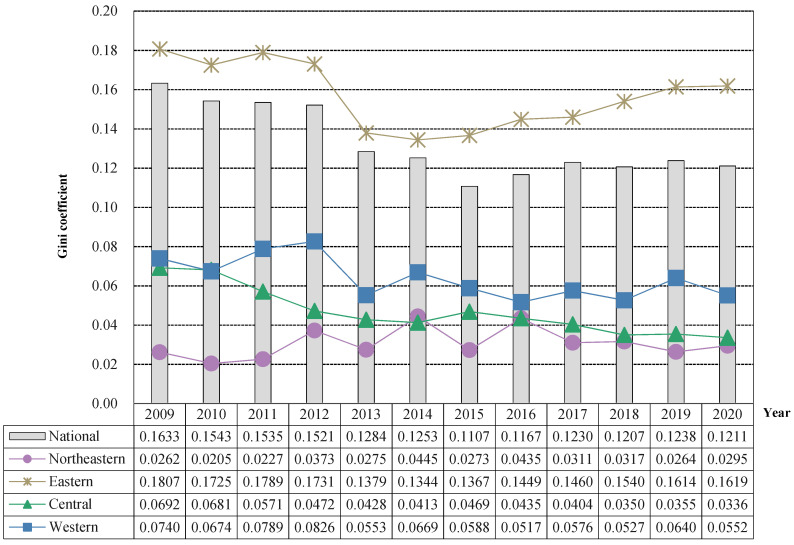
Intra-regional differences in public health level in China from 2009 to 2020.

**Figure 6 healthcare-11-01459-f006:**
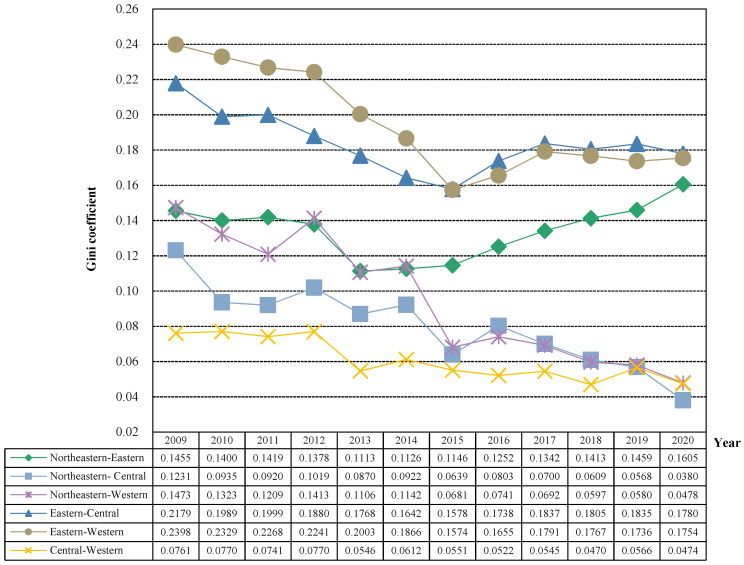
Inter-regional differences in public health level in China from 2009 to 2020.

**Figure 7 healthcare-11-01459-f007:**
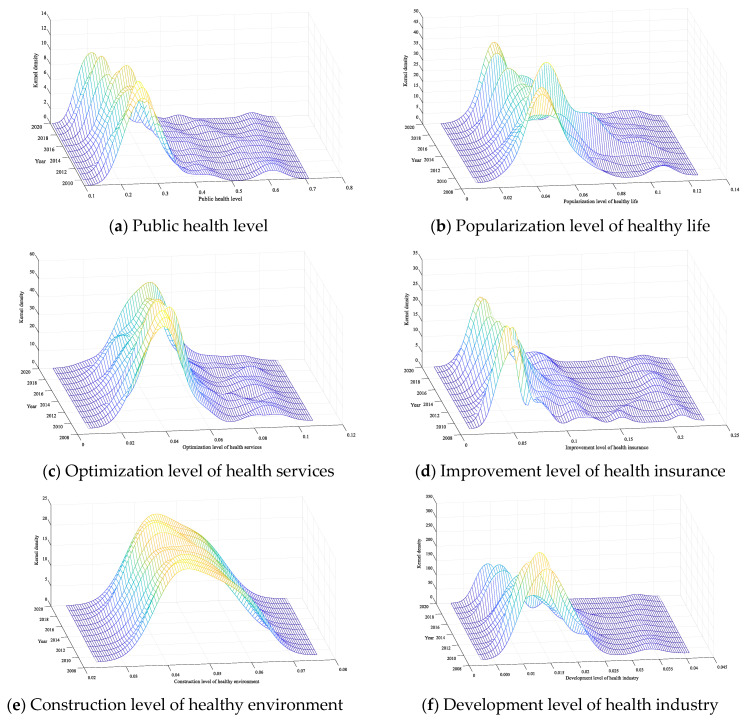
The dynamic evolution of public health and its sub-dimensions in China from 2009 to 2020.

**Figure 8 healthcare-11-01459-f008:**
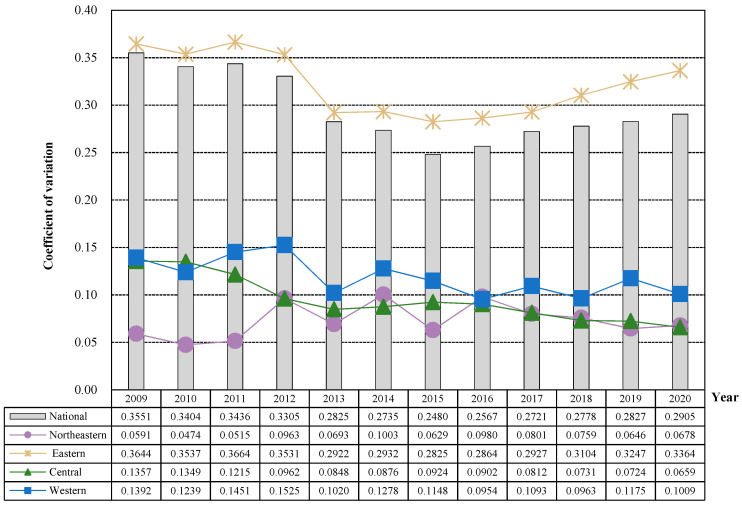
Coefficients of variation of public health levels in four major regions of China from 2009 to 2020.

**Table 1 healthcare-11-01459-t001:** Evaluation index system of China’s public health level.

Dimension Indicators	Element Indicators	Basic Indicators	Unit	Attribute
Popularization level of healthy living (0.1717)	Fundamentals of Healthy Living (0.0228)	Average life expectancy (0.0106)	year	+
		Average years of schooling (0.0067)	year	+
		Mortality rate of the population (0.0055)	%	−
	Healthy lifestyle (0.1489)	Cost of culture and tourism per capita (0.0638)	CYN	+
		Number of public health activities per 10,000 people (0.0619)	times/10,000 people	+
		Number of health and hygiene training per 10,000 people (0.0232)	times/10,000 people	+
Optimization level of health services (0.1198)	Provision of health services (0.0797)	Number of health personnel per 10,000 population (0.0191)	person	+
		Healthcare expenditure per capita (0.0342)	CYN	+
		Number of beds in health care facilities per 10,000 population (0.0264)	piece/10,000 people	+
	Demand for health services (0.0398)	Average length of hospital stay (0.0030)	day	−
		Maternal mortality (0.0016)	1/100,000	−
		Per capita number of consultations and treatments in medical and health institutions (0.0352)	person-times/person	+
Improvement level of health insurance (0.2867)	Basic health insurance (0.1514)	Health insurance participation rate (0.0651)	%	+
		Participation rate of maternity insurance (0.0400)	%	+
		Participation rate of work-related injury insurance (0.0463)	%	+
	Social health security (0.1353)	Proportion of health expenditure (0.0193)	%	+
		Public administration and social security unit density (0.0771)	pieces/km^2^	+
		Per capita expenditure on social medical assistance (0.0389)	%	+
Construction level of healthy environment (0.0996)	Construction of natural environment (0.0505)	Proportion of days with good air quality (0.0076)	%	+
		Forest coverage (0.0429)	%	+
	Construction of living environment (0.0159)	Green coverage in built-up areas (0.0054)	%	+
		Agricultural non-point source pollution index (0.0105)	/	−
	Construction of public safety (0.0332)	Number of deaths and injuries per 10,000 people in traffic accident (0.0115)	person	−
		Number of deaths and injuries per 10,000 people in traffic accident (0.0217)	%	+
Development level of health industry (0.3222)	Medical and pharmaceutical industry (0.1748)	Coverage of 10,000 medical and health institutions (0.0301)	pieces/10,000 people	+
		Healthcare institutions’ revenue as a percentage of GDP (0.0180)	%	+
		The main revenue of the pharmaceutical manufacturing industry accounts for the proportion of GDP (0.0361)	%	+
		Per capita has the amount of finished pharmaceutical products (0.0942)	CYN/person	+
	Healthcare and leisure industry (0.1474)	Number of nursing beds per 10,000 elderly population (0.0988)	pieces/10,000 people	+
		Number of cultural and sports leisure industries per 10,000 people (0.0486)	pieces/10,000 people	+

Note: The data in the table are indicator weights calculated using the entropy method. For details, please refer to the relevant calculation steps of the entropy method. In addition, + indicates that the indicator attribute is positive, − indicates that the indicator attribute is negative.

**Table 2 healthcare-11-01459-t002:** Sources and Contribution Rates of Difference in China’s Provincial Public Health Levels from 2009 to 2020.

Year	Intraregional Differences		Interregional Differences		Hypervariable Density	
	Differences	Contribution Rate (%)	Differences	Contribution Rate (%)	Differences	Contribution Rate (%)
2009	0.0359	21.97	0.1155	70.76	0.0119	7.26
2010	0.0338	21.90	0.1122	72.73	0.0083	5.37
2011	0.0357	23.25	0.1076	70.13	0.0102	6.62
2012	0.0349	22.94	0.1056	69.39	0.0117	7.67
2013	0.0266	20.73	0.0964	75.02	0.0055	4.26
2014	0.0276	22.00	0.0880	70.23	0.0097	7.77
2015	0.0268	24.19	0.0728	65.71	0.0112	10.10
2016	0.0270	23.18	0.0795	68.14	0.0101	8.68
2017	0.0280	22.76	0.0855	69.50	0.0095	7.74
2018	0.0282	23.38	0.0837	69.38	0.0087	7.24
2019	0.0306	24.72	0.0814	65.75	0.0118	9.53

**Table 3 healthcare-11-01459-t003:** Dynamic evolution of provincial public health level and its sub-dimensions in China from 2009 to 2020.

Dimension	Distribution Location	Distribution Pattern of Main Peak	Distribution Ductility	Wave Number
Public health level	Move right first and then left	The height fluctuation decreases and the width becomes larger	Right trailing, ductility convergence	single peak
Popularization level of healthy life	Move right first and then left	The height first decreases and then rises, and the width first becomes wide and then narrow	Right trailing, ductility convergence	single peak
Optimization level of health services	right shift	The height first decreases and then rises, and the width first becomes wide and then narrow	Right trailing, ductility convergence	single peak
Improvement level of health insurance	right shift	The height rises and the width becomes narrow	Right trailing, ductility convergence	single peak
Construction level of healthy environment	Basically unchanged	The height rises and the width becomes narrow	Right trailing, ductility convergence	single peak
Development level of health industry	right shift	The height drops and the width becomes larger	Right trailing, ductility convergence	single peak

**Table 4 healthcare-11-01459-t004:** Moran index of China’s public health level from 2009 to 2020.

Year	2009	2010	2011	2012	2013	2014	2015	2016	2017	2018	2019	2020
Moran index	0.296	0.291	0.216	0.233	0.292	0.257	0.253	0.246	0.269	0.252	0.217	0.211
Z value	3.020	2.961	2.569	2.525	3.196	2.806	2.717	2.747	2.888	3.067	2.592	2.765
P value	0.007	0.008	0.019	0.016	0.007	0.011	0.008	0.011	0.007	0.011	0.018	0.017

**Table 5 healthcare-11-01459-t005:** Absolute β convergence characteristics of public health level in China.

Region	National	Northeastern	Eastern	Central	Western
Model type	Bidirectional fixed SLM	Bidirectional fixed SDM	Bidirectional fixed SEM	Bidirectional fixed SDM	Bidirectional fixed SEM
β (lnPHL)	−0.3403 *** (−8.4813)	−0.2364 (−1.1907)	−0.3906 *** (−4.9151)	−0.5959 *** (−5.8469)	−0.3901 *** (−5.7272)
θ (w × lnPHL)		−0.9848 *** (−4.1054)		−0.5500 *** (−2.7911)	
ρ/λ	0.1254 * (1.8276)	−0.5760 *** (−6.4777)	−0.0984 (−0.9989)	−0.0340 (−0.2544)	0.1345 (1.1917)
$R^{2}$	0.4682	0.3170	0.4826	0.6907	0.4822
Log-likelihood	594.1604	79.5240	219.7158	142.8919	210.6880
Spatial fixation effect	79.9221 ***	14.8078 ***	27.7298 ***	14.5577 **	28.2654 ***
Time fixed effect	128.6659 ***	49.3858 ***	49.8167 ***	46.8261 ***	58.5051 ***
Hausman test	56.1236 ***	16.1047 ***	19.0650 ***	20.3696 ***	7.8915 **
LM spatial lag	52.9503 ***	6.9489 ***	2.5624	17.6795 ***	8.1489 ***
Robust LM spatial lag	2.3186	2.1735	6.4700 **	1.9288	1.6526
LM spatial error	50.8586 ***	8.0974 ***	3.8628 **	19.9913 ***	11.1405 ***
Robust LM spatial error	0.2269	3.3220 *	7.7705 ***	4.2406 **	4.6442 **
Wald test spatial lag	1.4283	16.8543 ***	3.3654 *	7.7903 ***	0.1576
LR test spatial lag	1.4483	18.7301 ***	2.8988 *	6.8731 ***	0.1697
Wald test spatial error	0.6377	2.9311 *	1.6080	7.9948 ***	0.2631
LR test spatial error	0.6249	2.7130 *	1.5984	7.5637 ***	0.2570
ν (%)	3.7815	2.4519	4.5025	8.2372	4.4951
Number of observations	341	33	110	66	132

Note: ***, ** and * represent significance at the level of 1%, 5% and 10%, respectively, and the values in brackets are t values.

**Table 6 healthcare-11-01459-t006:** Condition β convergence characteristics of public health level in China.

Region	National	Northeastern	Eastern	Central	Western
Model type	Bidirectional fixed SDM	Bidirectional fixed SEM	Bidirectional fixed SDM	Bidirectional fixed SDM	Bidirectional fixed SDM
β (lnPHL)	−0.4872 *** (−10.7469)	−0.4174 *** (−5.3425)	−0.6722 *** (−7.5332)	−0.5017 *** (−10.9154)	−0.5697 *** (−8.9458)
ln*PGDP*	0.0004 (1.1885)	0.0025 ** (2.1009)	0.0007 * (1.8469)	0.0046 *** (2.5902)	−0.0016 (−1.4804)
lnFP	−0.0209 *** (−2.9698)	−0.0953 *** (−2.9456)	−0.0333 (−0.5955)	0.1206 *** (2.9134)	−0.0069 (−0.6922)
lnUR	−0.0012 (−0.4535)	−0.0068 * (−1.8110)	0.0051 (1.5139)	0.0527 ** (2.2118)	−0.0036 (−0.4204)
lnPD	−0.3586 ** (−1.9713)	−0.0336 (−0.1003)	−0.5391 *** (−3.2636)	13.0568 *** (5.9767)	1.5571 (0.8119)
lnAII	−0.0017 (−1.3882)	−0.0022 (−0.8503)	−0.0018 (−0.5687)	−0.0036 (−1.4059)	−0.0015 (−0.6586)
lnTI	0.0007 (0.7587)	−0.0172 ** (−2.1849)	0.0022 ** (2.5120)	0.0035 (0.7911)	−0.0096 ** (−1.9952)
lnFDI	0.0027 (0.8289)	0.0026 (0.6931)	0.0002 (0.0364)	0.0346 ** (2.2390)	−0.0276 ** (−2.3133)
θ (w × lnPHL)	0.1498 (1.5950)		−0.3144 ** (−2.0685)	−1.1339 *** (−5.1960)	−0.2688 (−1.4586)
w × ln*PGDP*	0.0004 (0.7200)		0.0010 (1.4074)	0.0138 ** (2.4101)	−0.0032 (−0.9809)
*w* × ln*FP*	0.0049 (0.3412)		0.0969 (1.1420)	0.1889 *** (2.6484)	−0.0006 (−0.0288)
w × ln*UR*	0.0016 (0.2621)		0.0078 (1.2520)	0.1174** (2.4052)	0.0129 (0.5941)
w × ln*PD*	−0.5516 (−1.1691)		−0.7097 ** (−2.1592)	17.1791 *** (4.7317)	−9.0360 * (−1.7800)
w × ln*AII*	0.0077 *** (2.9678)		0.0090 * (1.8608)	−0.0080 (−1.1012)	0.0132 *** (2.9054)
w × ln*TI*	0.0006 (0.3740)		0.0030 * (1.8981)	0.0071 (0.6168)	−0.0092 (−0.7543)
w × ln*FDI*	0.0181 * (1.9459)		0.0224** (2.3470)	0.0692** (2.2079)	−0.0342 (−1.4459)
ρ/λ	0.1285 * (1.7806)	−0.3860 *** (−3.2609)	−0.2850 *** (−3.0255)	−0.2361* (−1.8010)	−0.2850 ** (−2.2549)
$R^{2}$	0.5451	0.5247	0.6639	0.8510	0.6376
Log-likelihood	620.805	90.3228	239.4853	163.0740	232.8869
Spatial fixation effect	85.8069 ***	2.3628	29.9298 ***	15.8962 **	37.6283 ***
Time fixation effect	120.3901 ***	47.0265 ***	56.7612 ***	43.6416 ***	47.9145 ***
Hausman test	94.2356 ***	27.2852 *	36.0469 ***	82.0590 ***	104.5503 ***
LM spatial lag	39.4155 ***	0.2942	4.2296 **	17.2606 ***	7.6863 ***
Robust LM spatial lag	2.0020	1.3584	3.7100 *	8.4494 ***	3.0086 *
LM spatial error	53.6632 ***	3.4368 *	3.8976 **	16.1164 ***	10.5398 ***
Robust LM spatial error	16.2496 ***	4.5010 **	5.0544 **	7.7856 ***	5.8620 **
Wald test spatial lag	19.5852 **	1.7894	19.9278 **	67.8000 ***	32.1672 ***
LR test spatial lag	19.2481 **	1.7019	17.2229 **	28.1844 ***	27.7114 ***
Wald test spatial error	18.7307 **	2.3170	15.4386 *	48.2942 ***	30.2700 ***
LR test spatial error	18.2029 **	2.2098	14.4953 *	29.3109 ***	27.1700 ***
ν (%)	6.0715	4.9114	10.1396	6.3323	7.6661
Number of observations	341	33	110	66	132

Note: ***, ** and * represent significance at the level of 1%, 5% and 10%, respectively, and the values in brackets are t values.

## Data Availability

Data available on request.

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
