# Peer review of "Regional Differences, Dynamic Evolution and Convergence of Public Health Level in China"

_healthcare, 2023, doi:10.3390/healthcare11101459_

Round 1

Reviewer 1 Report

Overall assessment

I consider the work to be well written, fluent in content and language, and clearly structured so that it is easy to read despite its considerable length.

Abstract

Line 28f An abstract should be understandable in itself, without reading the full text of the paper. Avoid mathematical symbols here, replace them with a verbal explanation, similar to the other parts of the abstract.

1. Introduction

Line 38ff The meaning of these figures remains unclear to me. An increase by a factor of 1000 or more within one decade seams very unlikely. One conclusion of the authors is that China’s public health system still needs to be improved today. If today’s system is the result of improvements by a factor of 1000 or more, this would mean in retrospect that China’s public health system was virtually non existent before 2009. Which seems very unlikely. Please clarify this in the paper.

Section 3.2 Research Methods

The methods used are described in successive sections. There are some inconsistencies in terms of consistency of terminology and mathematical symbols.

Examples:

In line 256 the symbol N stands for ‘year’; in line 316 the symbol N denotes ‘the number of regional cities’.

Some of the terms used do not seem to be consistent, sometimes the terms ‘provinces’ and ‘regions’ are used, sometimes it is ‘regional cities’, ‘cities’, ‘spatial units’, …

Please use consistent terminology throughout the paper for the spatial units under consideration to avoid confusion. Perhaps the terms ‘provinces’ and ‘regions’ could serve this purpose quite well.

Section 3.3 Data Collection and Pretreatment

The quality of the input data is decisive for the quality of the result of any analysis. The authors should add some details about the properties of the statistical input data used. For example, they claim, that ‘missing data … were complemented …’ In principle there is nothing wrong with that. But how much data was missing, what was the rate of completeness of the input data? How (in-)homogeneous are the data in terms of their spatial and temporal distribution? Are the same semantics of terms used throughout the data fot the entire study area? And so forth for all generally accepted aspects of data quality.

Section 4.1.2 Spatial Distribution

Figure 4. Spatial distribution of provincial public health levels in China from 2009 to 2020.

The class name ‘Lower level’ points to ‘lower than low’ and, therefore, in the wrong direction. Consider replacing the class names accordingly. The full text states ‘ … a transition from "low and low level continuous, medium level sporadic, and high level sporadic" to "low level disappearing, low and medium level continuous, and high level sporadic"’ and similar. How do these notations relate to the class names in Figure 4? It seems that there are some inconsistencies here.

Section 4.4.1 Spatial Convergence of the Public Health Level in China

The reference of the first sentence remains unclear. If the sentence refers to Figure 8, than that should be mentioned ‘As Figure 8 shows …’

5. Conclusions and Policy Suggestions

Section 5.1. is rather an executive summary than a conclusion. Essentially, you repeat the summarized results of your research. But what do you conclude from these findings?

Section 5.2, part 3. Although this is a highly reasonable recommendation, I can not see how it is supported by the research findings. Therefore, I tend to suggest removing this part. But I am divided here.

Minor issues

Line 147 ‘Maternal mortality rate’ is mentioned twice, please correct.

Line 292 should it not read ‘region j and the r province in region h’ instead of ‘region h and the r province in region h’?

Line 317 should read ‘Formula (12)’ instead of ‘Formula (17)’.

Line 456ff ‘low level’ is mentioned twice, please correct. Something is wrong with the sentence ‘… and select 2009, In 2012 …’ (line 459), please correct.

The term ‘crown epidemic’ occurs several times. What is meant by that? COVID epidemic? Please correct.

A non-binding recommendation. Using the same scale for all four Subfigures 3(a), 3(b), 3(c), 3(d) would make the differences between the four regions more visible.

Author Response

Response to Reviewer 1 Comments

Dear Reviewer,

Thanks for your comments concerning our manuscript entitled “Regional Differences, Dynamic Evolution and Convergence of Public Health Level in China” (ID: healthcare-2347533). According to these suggestions, we have made several modifications in this paper. Our responses to each suggestion for revision are as follows:

Point 1: Abstract

Line 28f An abstract should be understandable in itself, without reading the full text of the paper. Avoid mathematical symbols here, replace them with a verbal explanation, similar to the other parts of the abstract.

Revision: Thank you for the feedback from the reviewer. We have made revisions to the 28th line of the abstract section based on the reviewers' comments and have marked it in a revision mode in the original manuscript. The specific modifications are as follows:

Finally, the level of public health in China has a significant spatial correlation. Except for the northeast region, the growth rate of low level public health provinces in China and the other three major regions is higher than that of high level public health provinces, showing a certain convergence trend.

Point 2: Introduction

Line 38ff The meaning of these figures remains unclear to me. An increase by a factor of 1000 or more within one decade seams very unlikely. One conclusion of the authors is that China’s public health system still needs to be improved today. If today’s system is the result of improvements by a factor of 1000 or more, this would mean in retrospect that China’s public health system was virtually nonexistent before 2009. Which seems very unlikely. Please clarify this in the paper.

Reply: The number on line 38 mainly expresses that as of the end of 2021, China has invested a large amount in the absolute values of medical and health indicators such as public health expenditure, number of medical and health institutions, number of beds in medical and health institutions, and number of personnel in medical and health institutions. Compared with 2009 when the new medical reform began, it has increased by 4.792 times, 1.125 times, 2.140 times, and 1.797 times, respectively, making significant progress. Over the past decade, the representative indicators of public health in China have shown an increasing trend, with the highest increasing by 4.792 times and the lowest by 1.125 times. There is no evaluation expert's suggestion of 1000 times or more, and the decimal point may bring some misunderstanding to the evaluation experts.

Point 3: Section 3.2 Research Methods

The methods used are described in successive sections. There are some inconsistencies in terms of consistency of terminology and mathematical symbols.

Revision: Thank you for your careful review. We have checked and modified the consistency of the full text's terms and mathematical symbols item by item, and the specific modifications have been marked in the original manuscript.

Examples:

In line 256 the symbol N stands for ‘year’; in line 316 the symbol N denotes ‘the number of regional cities’.

Revision: Thank you for your careful review. In order to maintain the consistency of the full text terms and mathematical symbols, we have modified the symbol N stands for ‘year’ in 256 lines to the symbol M stands for ‘year’ according to the opinions of the review experts. In line 316 the symbol N denotes ‘the number of regional cities’ is modified to the symbol N denotes ‘the number of regional provinces’ and marked in revised mode in the original manuscript.

Some of the terms used do not seem to be consistent, sometimes the terms ‘provinces’ and ‘regions’ are used, sometimes it is ‘regional cities’, ‘cities’, ‘spatial units’, …

Revision: Thank you for your careful review. There are indeed some inconsistencies in terms in the article. We have adjusted the ‘regional cities’, ‘cities’, and ‘spatial units’ in the original manuscript to regions and provinces according to expert opinions, and have marked them in a revised mode in the article. Specific modifications are as follows:

(1) modify the ‘region’ of line 255 to ‘province’.

(2) modify the ‘evaluation units’ in line 265 to the ‘evaluation province’.

(3) modify the ‘regional cities’ in line 320 to the ‘regional provinces’.

(4) modify the ‘city’ in line 370 to the ‘province’.

………..

In addition, other details of this section have been modified and highlighted in modification mode.

Please use consistent terminology throughout the paper for the spatial units under consideration to avoid confusion. Perhaps the terms ‘provinces’ and ‘regions’ could serve this purpose quite well.

Reply: Thank you for the suggestions from the reviewer. We have carefully checked the original manuscript and, according to the reviewers' opinions, have unified it using ‘provinces’ and ‘regions’.

Point 4: Section 3.3 Data Collection and Pretreatment

The quality of the input data is decisive for the quality of the result of any analysis. The authors should add some details about the properties of the statistical input data used. For example, they claim, that ‘missing data … were complemented …’ In principle there is nothing wrong with that. But how much data was missing, what was the rate of completeness of the input data? How (in-)homogeneous are the data in terms of their spatial and temporal distribution? Are the same semantics of terms used throughout the data fot the entire study area? And so forth for all generally accepted aspects of data quality.

Reply: Thank you for the feedback from the reviewer. This article aims to measure the level of public health in China since the new healthcare reform, and examine its spatio-temporal distribution, regional differences, dynamic evolution, and convergence characteristics. The research sample consists of 31 provinces (municipalities, autonomous regions) in China from 2009 to 2020. To accurately assess the level of public health in China, we extensively consider the reliability, continuity, and comparability of data. Supplementary explanations for missing data are as follows:

Firstly, considering the comparability of the data, this article did not include Hong Kong, Macao, and Taiwan in the evaluation samples. From the perspective of data measurement, consistency in statistical caliber is necessary to ensure comparability of results. But the fact is that China adopts one country, two systems, and the statistical caliber of Hong Kong, Macao, and Taiwan is not consistent with that of the mainland, making the results not comparable. Therefore, in order to achieve unified and standardized statistical data in terms of statistical scope and methods during the sample period, and ensure comparability of the data within the system's usage range, this article excluded Hong Kong, Macao, and Taiwan, and used 31 provinces (municipalities directly under the central government, autonomous regions).

Secondly, there are also some missing values in data acquisition. This article uses 30 indicators to measure the public health level of 31 provinces and cities in China from 2009 to 2020, with a total of 11160 data values. Among them, the missing data indicators include Mortality rate of the population, Number of public health activities per 10000 people, Number of health and hydrogen training per 10000 people, with missing values and sample sizes of 6/372, 8/372, and 4/372, respectively; The deletion rates were 1.61%, 2.15%, and 1.07% (Table 1). There are a total of 18 missing data values, accounting for 0.16%. Whether it is a total indicator or a single indicator, the average data missing rate is within 3%. From the overall situation of the dataset, the proportion of missing values is relatively small, and the information loss is limited. From the distribution of missing values, only three indicators have a small amount of missing values and do not have obvious distribution patterns or features. Completing them does not affect the expression of feature information or inaccurate model fitting.

Thirdly, in order to ensure the balance between the long-term time distribution and the cross-sectional distribution, we use the linear function method (TREND function) to fill in the missing data.

Table 1. Missing data of indicators for China's public health level

Dimension indicators

Element indicators

Basic indicators

sample size

Missing data

Popularization level

of healthy living

Fundamentals of Healthy Living

Average life expectancy

372

0

Average years of schooling

372

0

Mortality rate of the population

372

6

(1.61%)

Healthy lifestyle

Cost of culture and tourism per capita

372

0

Number of public health activities per 10,000 people

372

8

(2.15%)

Number of health and hygiene training per 10,000 people

372

4

(1.07%)

Optimization level

of health services

Provision of health services

Number of health personnel per 10,000 population

372

0

Health care expenditure per capita

372

0

Number of beds in health care facilities per 10,000 population

372

0

Demand for health services

Average length of hospital stay

372

0

Maternal mortality

372

0

Per capita number of consultations and treatments in medical and health institutions

372

0

Improvement level

of health insurance

Basic health insurance

Health insurance participation rate

372

0

Participation rate of maternity insurance

372

0

Participation rate of work-related injury insurance

372

0

Social health security

Proportion of health expenditure

372

0

Public administration and social security unit density

372

0

Per capita expenditure on social medical assistance

372

0

Construction level

of healthy environment

Construction of natural environment

Proportion of days with good air quality

372

0

forest coverage

372

0

Construction of living environment

Green coverage in built-up areas

372

0

Agricultural non-point source pollution index

372

0

Construction of public safety

Number of deaths and injuries per 10,000 people in traffic accident

372

0

Number of deaths and injuries per 10,000 people in traffic accident

372

0

Development level

of health industry

Medical and pharmaceutical industry

Coverage of 10,000 medical and health institutions

372

0

Healthcare institutions' revenue as a percentage of GDP

372

0

The main revenue of the pharmaceutical manufacturing industry accounts for the proportion of GDP

372

0

Per capita has the amount of finished pharmaceutical products

372

0

Healthcare and leisure industry

Number of nursing beds per 10,000 elderly population

372

0

Number of cultural and sports leisure industries per 10,000 people

372

0

Total

11160

18

(0.16%)

Point 5: Spatial Distribution

Figure 4. Spatial distribution of provincial public health levels in China from 2009 to 2020.

The class name ‘Lower level’ points to ‘lower than low’ and, therefore, in the wrong direction. Consider replacing the class names accordingly.

Revision: Thank you for your careful review. Based on the opinions of the reviewers, we have made modifications to Figure 4, changing the four levels from ‘Low Level, Lower Level, Medium Level, and High Level’ to ‘lowest Level, Low Level, Medium Level, and High Level’, and have marked them in revision mode in the original manuscript. The specific modified map is as follows:

The full text states ‘ … a transition from "low and low level continuous, medium level sporadic, and high level sporadic" to "low level disappearing, low and medium level continuous, and high level sporadic"’ and similar. How do these notations relate to the class names in Figure 4? It seems that there are some inconsistencies here.

Revision: Regarding the full text states you pointed out '... a transition from "low and low level continuous, medium level sporadic, and high level sporadic" to "low level disappearing, low and medium level continuous, and high level sporadic"’ and similar. We have also made revisions to the issue of inconsistency with the classification in Figure 4 based on reviewer opinions, and have indicated it in a revision mode in the original text.

The specific modifications are as follows: from a specific province perspective, since the new medical reform, the public health level in China has undergone a transformation from "lowest level and low level contiguous, medium level sporadic, and high level none" to "lowest level disappearing, low level and medium level contiguous, and high level sporadic", showing a good overall development trend.

Point 6: Conclusions and Policy Suggestions

Section 5.1. is rather an executive summary than a conclusion. Essentially, you repeat the summarized results of your research. But what do you conclude from these findings?

Revision: Thank you for your review suggestions. This manuscript has been re summarized based on the opinions of the review experts and marked in a revised mode in the original manuscript. The specific modifications are as follows:

Firstly, the overall level of public health in China is relatively low, and there is a significant imbalance in regional development, presenting a spatial distribution pattern of "high in the east and low in the central and western regions". Further research has found that the structural issues in China's public health level are prominent, mainly reflected in the development of the health industry and the optimization of health services, which are the core driving forces for promoting the improvement of China's public health level since the new medical reform. The improvement of health insurance and the popularization of healthy life are the backbone of China's public health improvement. The construction of a healthy environment has become a shortcoming that hinders the improvement of China's public health level since the new medical reform.

Secondly, the overall regional differences in China's public health level show a "V-shaped" downward trend of first decreasing and then increasing. Among them, the inter-regional differences are the main sources of the overall regional differences in China's public health level, and the differences between the northeastern and eastern regions are expanding. In addition, the contribution rate of intra-regional differences to the overall regional differences in China's public health is on the rise, and special attention needs to be paid to prevent the expansion of the overall regional differences in China's public health level.

Third, except that the construction of a healthy environment remains basically unchanged, China's public health level and its sub-dimensions are on an upward trajectory (excluded by the impact of the COVID-19 pandemic in 2020), and there is no polarization. However, there are some provinces within the region that have significantly higher levels of public health and their sub-dimensions than other provinces in the unified region, and the overall level of public health in China and the degree of dispersion of the sub dimensions of health industry development are on the rise.

Fourthly, from the perspective of convergence, except for the northeast region, the public health levels of the whole country and the other three major regions will show a certain convergence trend over time. From the perspective of  convergence, without considering economic and social factors, its convergence characteristics are basically consistent with convergence, while the convergence characteristics in northeast China are not significant, indicating the possibility of regional differences expanding. But after considering economic and social factors, the public health level across the country and the four major regions shows a convergence trend. In addition, the impact of economic development level, financial pressure, urbanization level, population density, advanced industrial structure, scientific and technological innovation level and opening up level on the convergence of public health level in the four major regions is significantly heterogeneous.

Section 5.2, part 3. Although this is a highly reasonable recommendation, I can not see how it is supported by the research findings. Therefore, I tend to suggest removing this part. But I am divided here.

Revision: Thank you for your suggestion. According to the opinions of the reviewer, Section 5.2 and Part 3 have been deleted from this manuscript and marked in revised mode in the original manuscript. Thank you again for the careful review and relevant suggestions from the reviewer.

Point 7: Minor issues

Line 147 ‘Maternal mortality rate’ is mentioned twice, please correct.

Revision: Thank you for your careful review. The first mention of 'Material morality rate' in Line 147 has been corrected to ' Neonatal mortality rate' and marked in revision mode in the original manuscript.

Line 292 should it not read ‘region j and the r province in region h’ instead of ‘region h and the r province in region h’?

Revision: Thank you for your careful review. We have revised the 'region h and the r provision in region h' of Line 292 to 'region j and the r provision in region h' according to your feedback, and have marked it in revision mode in the original manuscript.

Line 317 should read ‘Formula (12)’ instead of ‘Formula (17)’.

Revision: Thank you for your careful review. We have revised the 'Formula (17)' of Line 317 to 'Formula (12)' according to your feedback, and have marked it in revision mode in the original manuscript.

Line 456ff ‘low level’ is mentioned twice, please correct. Something is wrong with the sentence ‘… and select 2009, In 2012 …’ (line 459), please correct.

Revision: Thank you for your careful review. We have made modifications to Figure 4 based on the above comments, changing the first 'low level' of Line 460 to 'lowest level'. In addition, corrections have been made to the sentences in line 463, and specific modifications have been annotated in revised mode in the original manuscript.

The term ‘crown epidemic’ occurs several times. What is meant by that? COVID epidemic? Please correct.

Revision: Thank you for your careful review. We have carefully reviewed and revised the entire text, and have revised all ‘crown epidemic’ in the original manuscript to ‘COVID-19 epidemic’, all of which have been marked in revision mode. Specifically modified as: Line 70, Line 437, Line 477, Line 630, and Line 859.

A non-binding recommendation. Using the same scale for all four Subfigures 3(a), 3(b), 3(c), 3(d) would make the differences between the four regions more visible.

Revision: Thank you for your feedback. In order to increase comparability between regions, we have unified the scales of the three sub-figures 3 (a), 3 (b), 3 (c), and 3 (d) of Figure 3, and marked them in revision mode in the original manuscript. The specific modifications are shown in the following figure:

Reviewer 2 Report

The authors of the article have attempted to present important research issues concerning the level of public health in China. In the health care system, it is an important issue from the perspective of the functioning of the state. Nowadays, the health policy of most countries is based on the view that the health of the population is a necessary condition for economic efficiency and prosperity. Therefore, the subject matter taken up by the authors is part of the research trend of selected multi-faceted and multi-directional actions taken to protect health and combat disease threats of epidemic and pandemic nature. After getting acquainted with the content of the study, I suggest the authors, improve the study, and supplement the content in the subsections. Below are detailed comments on the article:

1. I suggest supplementing the abstract with the purpose of the article.

2. In the Introduction chapter, please consider how the authors' research has broadened the knowledge about the problem.

3. In the Literature Review chapter Introduction, please supplement with research hypotheses, which should then be confirmed/rejected in the Results chapter based on the conducted research.

4. Under Table 1 (L218-219) a legend should be added explaining the characters used, e.g. "+" and "-".

5. Subchapter 3.3 Data Collection and Pretreatment (L389-400) should be supplemented with the procedure for collecting materials, description of data search criteria, date of data collection, etc.

6. The readability of drawings should be improved 7 Some elements are illegible, e.g. scale (L670)

7. Based on the global Moran statistics, maps can be generated.

8. In the Research Methods chapter, complete the information on the statistical package in which the individual statistical calculations were performed.

9. Chapter 5 5. Conclusion and Policy Suggestions should be supplemented with further research plans related to the analyzed issues.

Author Response

Response to Reviewer 2 Comments

Dear Reviewer,

Thanks for your comments concerning our manuscript entitled “Regional Differences, Dynamic Evolution and Convergence of Public Health Level in China” (ID: healthcare-2347533). According to these suggestions, we have made several modifications in this paper. Our responses to each suggestion for revision are as follows:

Point 1: I suggest supplementing the abstract with the purpose of the article.

Revision: Thank you for your careful review. We have added the research purpose of this manuscript in the abstract according to your feedback, and have annotated it in revised mode in the original manuscript. The specific content is:

People's health is a necessary condition for the country's prosperity. Under the background of the COVID-19 pandemic and frequent natural disasters, exploring the spatial and temporal distribution, regional differences and convergence of China's provincial public health level is of great significance to promote the coordinated development of China's regional public health and achieve the strategic goal of "healthy China".

Point 2: In the Introduction chapter, please consider how the authors' research has broadened the knowledge about the problem.

Revision: Thank you for your suggestion. We have added the innovative points of this study in the introduction section based on your feedback, and have annotated them in a revised mode in the original manuscript. The supplementary content is as follows:

Compared with existing research, this manuscript mainly expands public health related research from the following aspects: In terms of research content, this study breaks through previous studies that only focus on the spatiotemporal distribution and influencing factors of public health at a single level. Using Dagum Gini coefficient, Kernel density function, and spatial econometric models, it explores the basic laws of China's public health level from multiple levels such as regional differences, dynamic evolution, and convergence. In terms of research methods, this study considered the spatial spillover effect of public health levels, corrected the strict assumptions of traditional econometric models, and ensured the accuracy of the calculation results. In terms of indicator design, this study combines the actual situation in China with the guiding ideology and strategic goals of the "Healthy China 2030" Plan Outline, and constructs comprehensive indicators including healthy living, health services, health security, health environment, and health industry. This enriches the indicator system for public health evaluation and corrects the deviation of previous single indicator measurement.

Point 3: In the Literature Review chapter Introduction, please supplement with research hypotheses, which should then be confirmed/rejected in the Results chapter based on the conducted research.

Revision: Thank you for your suggestion. Based on your feedback, we have added relevant research hypotheses in the literature review section and annotated them in a revised format in the original manuscript. The supplementary content is as follows:

China has a vast territory, with uneven resource endowments and economic and social development in various regions, and significant regional differences in public health levels. The eastern region has been affected by the reform and opening up policy, and the process of industrialization and urbanization has rapidly advanced, attracting a large number of medical and health talents to gather here, establishing a relatively complete medical and health system, and the overall level of regional public health is relatively high. The central and western regions are located inland, with scarce resources and limited information. The level of economic and social development is not high, and they have long faced the problem of "difficult employment and retention". Medical and health resources are scarce, the public health system is not sound, and the overall level of public health in the region is relatively low. Based on the above analysis, hypothesis 1 of this article is proposed: there are significant regional differences in China's public health level, and inter-regional differences are the main source.

In addition, according to the first law of geography, everything within a spatial range is related, and if the distance is different, the interaction between the two also varies significantly. This spatial interaction is also understood as a spatial effect. This spatial effect can be divided into spatial dependence and spatial heterogeneity. Spatial dependence mainly refers to the fact that individuals in space are not independent of each other, but rather interconnected, which is mainly caused by the spillover of factors, technologies, and policies between regions. Spatial heterogeneity is due to different geographical locations and natural resource conditions, leading to certain differences between regions, such as coastal and inland, southern and northern, eastern and western, etc. However, existing studies on the evaluation of public health levels often use traditional econometric models, which assume that individuals exist completely independently in space, do not comply with the first law of geography, and there is a certain computational bias. This study starts from the actual situation in China, based on the guiding ideology and strategic goals of the "Healthy China 2030" Plan issued by the Central Committee of the Communist Party of China and the State Council, and draws on the reasonable parts of existing research, attempting to construct a Chinese public health evaluation index system from five dimensions: popularizing healthy life, optimizing health services, improving health insurance, constructing a healthy environment, and developing health industry. This evaluation index includes various levels of economy, society, ecology, and involves the flow of various resource elements, therefore, there is a significant spatial correlation. In addition, the convergence theory of neoclassical economics suggests that under the condition of diminishing marginal utility of capital in various regions, the growth rate of economically underdeveloped regions is higher than that of economically developed regions. With the promotion of technology, this gap continues to decrease over time, and the economic development level of each region is ultimately in a balanced state. Therefore, the convergence theory can also be applied to the development process of public health in China, where low level provinces of public health will gradually narrow the gap with high level provinces of public health under the combined effect of technology and policies, presenting a convergence characteristic. Based on the above analysis, hypothesis 2 of this paper is proposed: there is a significant spatial correlation in China's public health level, and it shows a certain convergence trend over time.

Point 4: Under Table 1 (L218-219) a legend should be added explaining the characters used, e.g. "+" and "-".

Revision: Thank you for your careful review. We have explained the meanings of the "+" and "-" used in the comments section of Table 1, and marked them in revision mode in the original manuscript. Specific modifications: "In addition, + indicates that the indicator attribute is positive, - indicates that the indicator attribute is negative.

Point 5: Subchapter 3.3 Data Collection and Pretreatment (L389-400) should be supplemented with the procedure for collecting materials, description of data search criteria, date of data collection, etc.

Reply: Thank you for your feedback. In conjunction with Reviewer 1, we have provided additional information on the data collection and processing process:

Firstly, considering the comparability of the data, this article did not include Hong Kong, Macao, and Taiwan in the evaluation samples. From the perspective of data measurement, consistency in statistical caliber is necessary to ensure comparability of results. But the fact is that China adopts one country, two systems, and the statistical caliber of Hong Kong, Macao, and Taiwan is not consistent with that of the mainland, making the results not comparable. Therefore, in order to achieve unified and standardized statistical data in terms of statistical scope and methods during the sample period, and ensure comparability of the data within the system's usage range, this article excluded Hong Kong, Macao, and Taiwan, and used 31 provinces (municipalities directly under the central government, autonomous regions).

Secondly, from the perspective of data sources, considering that the data collected and published by government agencies can ensure the comparability, reliability, scientific accuracy, and continuity of the data as much as possible, we choose to obtain data from government officials. The data used in this article are all from public data from various national departments and provinces from 2010 to 2021. Specifically, the data comes from the "China Health Statistical Yearbook" of the National Health Commission from 2010 to 2021, the "China Statistical Yearbook" of the National Bureau of Statistics, the "China Industrial Statistical Yearbook" of the Industrial Statistics Department of the National Bureau of Statistics, and the "China Basic Unit Statistical Yearbook" of the Census Center of the National Bureau of Statistics The China Education Statistical Yearbook Committee's "China Education Statistical Yearbook" as well as historical statistical yearbooks and statistical bulletins from various provincial statistical bureaus. From the perspective of data acquisition, we have been paying long-term attention to the publication of data and obtaining it in a timely manner. The data used in this article have been published for a long time by the National Bureau of Statistics and provincial statistical bureaus. We have been focusing on this field for a long time and are committed to establishing a Chinese public health database. In 2019, we began collecting data on the above platforms and obtained all the required data in 2022.

Thirdly, in order to ensure the balance between the long-term time distribution and the cross-sectional distribution, we use the linear function method (TREND function) to fill in the missing data. After completing the data, the statistical description is shown in Table 2.

Table 2. Descriptive Statistics of Evaluation Indicators for Public Health Level in China

indicators

Mean

Std

Min

Max

Obs

Average life expectancy

76.249

2.697

67.790

82.550

372

Average years of schooling

9.008

1.140

4.222

12.681

372

Mortality rate of the population

6.074

0.823

4.210

9.470

372

Cost of culture and tourism per capita

60.094

47.278

9.600

309.710

372

Number of public health activities per 10,000 people

0.584

0.562

0.002

3.102

372

Number of health and hygiene training per 10,000 people

7.548

3.625

0.182

19.357

372

Number of health personnel per 10,000 population

78.624

17.553

40.675

159.007

372

Health care expenditure per capita

1059.335

541.210

185.460

3110.400

372

Number of beds in health care facilities per 10,000 population

49.708

12.259

26.048

80.676

372

Average length of hospital stay

9.846

1.057

7.700

16.200

372

Maternal mortality

18.539

23.579

1.100

232.230

372

Per capita number of consultations and treatments in medical and health institutions

5.176

1.7616

2.512

11.364

372

Health insurance participation rate

21.731

13.470

7.618

79.561

372

Participation rate of maternity insurance

12.703

8.646

1.133

61.265

372

Participation rate of work-related injury insurance

15.347

9.889

2.821

57.891

372

Proportion of health expenditure

7.419

1.519

3.972

12.078

372

Public administration and social security unit density

3.971

4.010

0.089

21.802

372

Per capita expenditure on social medical assistance

391.057

227.694

67.753

1874.597

372

Proportion of days with good air quality

76.870

17.252

13.388

100.000

372

Forest coverage

32.104

17.911

4.020

66.800

372

Green coverage in built-up areas

39.001

4.147

18.100

49

372

Agricultural non-point source pollution index

95.625

72.461

1.962

334.491

372

Number of deaths and injuries per 10,000 people in traffic accident

2.367

1.154

0.359

6.004

372

Number of deaths and injuries per 10,000 people in traffic accident

5.674

1.128

3.478

12.430

372

Coverage of 10,000 medical and health institutions

7.533

3.265

2.012

21.366

372

Healthcare institutions' revenue as a percentage of GDP

3.358

0.8440

1.425

6.292

372

The main revenue of the pharmaceutical manufacturing industry accounts for the proportion of GDP

2.676

1.709

0.175

12.526

372

Per capita has the amount of finished pharmaceutical products

99.792

115.218

7.937

837.643

372

Number of nursing beds per 10,000 elderly population

2.458

3.113

0.314

37.721

372

Number of cultural and sports leisure industries per 10,000 people

209.653

136.738

3.900

583.200

372

Point 6: The readability of drawings should be improved 7 Some elements are illegible, e.g. scale (L670)

Revision: Thank you for your careful review. In order to increase the readability of Figure 7, we have made adjustments to the font and annotated it in revision mode in the original manuscript. The specific modifications are shown in the following figure:

(a) Public health level

(b) Popularization level of healthy life

(c) Optimization level of health services

(d) Improvement level of health insurance

(e) Construction level of healthy environment

(f) Development level of health industry

Figure 7. The dynamic evolution of public health and its sub-dimensions in China from 2009 to 2020.

Point 7: Based on the global Moran statistics, maps can be generated.

Reply: Thank you for your suggestion. As this article uses the global Moran Index and you pointed out that generating maps based on Moran statistical data is within the scope of local Moran Index research, this study mainly focuses on the global Moran Index. Therefore, there is no need to draw LISA clustering maps. According to the opinions of the evaluation experts and limited by space, we have drawn the Moran scatter plot for 2020 as follows:

Point 8: In the Research Methods chapter, complete the information on the statistical package in which the individual statistical calculations were performed.

Revision: Thank you for your suggestions. We have added the statistical packages used for each research method in the Research Methods chapter and marked them in a revision mode in the original text. The specific modifications are as follows:

  • Line 257 of 3.2.1. Global entropy method added "Stata 17.0 software"
  • Line 295 of 3.2.2. Dagum Gini Coefficient added “This paper uses MATLAB 2018a software for calculation.”
  • Line 312 of 3.2.3 Kernel density estimation added “This paper uses MATLAB 2018a software for calculation.”
  • Line 332 of 3.2.4. Global spatial autocorrelation added “This paper uses GeoDa software for calculation.”
  • Line 343 of 3.2.5. Convergence Model added “this paper uses MATLAB 2018a software and…”

Point 9: Chapter 5 5. Conclusion and Policy Suggestions should be supplemented with further research plans related to the analyzed issues.

Revision: Thank you for your suggestion. We have added the limitations of this study and future research plans based on your feedback at the end of the manuscript, and have annotated them in a revised format in the original manuscript. The specific supplementary content is:

In addition, this study has certain limitations. On the one hand, due to the vast area of most provinces, there are significant differences in the basic conditions for the development of internal public health levels. Therefore, there are still certain shortcomings in examining the regional differences, dynamic evolution, and convergence of China's public health level only at the provincial level. In the future, more effective micro data can be obtained through questionnaire surveys and interviews at smaller spatial scales such as cities, counties, and villages, providing data support for more targeted promotion of China's public health level. On the other hand, this study only attempted to preliminarily explore the basic characteristics of regional differences, dynamic evolution, and convergence of provincial public health levels in China. In the future, Mediation effect, difference in difference and other methods can be used to further explore the deep impact mechanism and policy net effect of the improvement of China's public health level, so as to improve the accuracy of the assessment.

Reviewer 3 Report

This study evaluates China's public health systems using multiple analytic approaches. I found this study presents a rigorous evaluation framework for national public health systems. And the empirical approaches are sound. However, it is not clear enough to state why this study is so important and how critical the results of this study can provide contributions. This research motivation could be much elaborated in the introduction section with clear research questions.

Based on the research questions, hypotheses could be provided. The hypotheses constitutes the main arguments and the research focus of this study. Also, with the hypotheses and their rationales, the authors can legitimate why this study considers various approaches to evaluate the public health systems. That means, the hypotheses are also developed to show how the hypotheses are examined.

Taken together, the authors may want to provide research questions in the introduction section, develop hypotheses by creating a new section for hypotheses, and elaborate why the empirical methods are designed in the methodology section.

Hope these comments help.

No issues here.

Author Response

Response to Reviewer 3 Comments

Dear Reviewer,

Thanks for your comments concerning our manuscript entitled “Regional Differences, Dynamic Evolution and Convergence of Public Health Level in China” (ID: healthcare-2347533). According to these suggestions, we have made several modifications in this paper. Our responses to each suggestion for revision are as follows:

Point 1: This study evaluates China's public health systems using multiple analytic approaches. I found this study presents a rigorous evaluation framework for national public health systems. And the empirical approaches are sound. However, it is not clear enough to state why this study is so important and how critical the results of this study can provide contributions. This research motivation could be much elaborated in the introduction section with clear research questions.

Reply: Thank you for your careful review. Firstly, the research question of this article is: How to design an evaluation index system for China's public health level? What are its spatiotemporal distribution patterns, regional differences, main sources, dynamic evolution characteristics, and convergence? The question form has been proposed in the introduction section. Secondly, the importance of this study includes two parts: practical significance and theoretical significance. The practical significance is that people's health is a sign of national rejuvenation and national prosperity. In this era of COVID-19 pandemic and frequent natural disasters, designing a scientific and reasonable evaluation index system to measure China's provincial public health level, and using a variety of methods to explore its regional differences and main sources, dynamic evolution characteristics and convergence can provide empirical support for government departments to formulate scientific policy plans. The theoretical significance is mainly elaborated in the introduction section based on the opinions of Reviewer 2, and the specific supplementary content is:

Compared with existing research, this manuscript mainly expands public health related research from the following aspects: In terms of research content, this study breaks through previous studies that only focus on the spatiotemporal distribution and influencing factors of public health at a single level. Using Dagum Gini coefficient, Kernel density function, and spatial econometric models, it explores the basic laws of China's public health level from multiple levels such as regional differences, dynamic evolution, and convergence. In terms of research methods, this study considered the spatial spillover effect of public health levels, corrected the strict assumptions of traditional econometric models, and ensured the accuracy of the calculation results. In terms of indicator design, this study combines the actual situation in China with the guiding ideology and strategic goals of the "Healthy China 2030" Plan Outline, and constructs comprehensive indicators including healthy living, health services, health security, health environment, and health industry. This enriches the indicator system for public health evaluation and corrects the deviation of previous single indicator measurement.

Point 2: Based on the research questions, hypotheses could be provided. The hypotheses constitutes the main arguments and the research focus of this study. Also, with the hypotheses and their rationales, the authors can legitimate why this study considers various approaches to evaluate the public health systems. That means, the hypotheses are also developed to show how the hypotheses are examined.

Revision: Thank you for your suggestion. Based on the comments of Reviewer 2, we have added relevant research hypotheses in the literature review section and annotated them in a revised format in the original manuscript. The supplementary content is as follows:

China has a vast territory, with uneven resource endowments and economic and social development in various regions, and significant regional differences in public health levels. The eastern region has been affected by the reform and opening up policy, and the process of industrialization and urbanization has rapidly advanced, attracting a large number of medical and health talents to gather here, establishing a relatively complete medical and health system, and the overall level of regional public health is relatively high. The central and western regions are located inland, with scarce resources and limited information. The level of economic and social development is not high, and they have long faced the problem of "difficult employment and retention". Medical and health resources are scarce, the public health system is not sound, and the overall level of public health in the region is relatively low. Based on the above analysis, hypothesis 1 of this article is proposed: there are significant regional differences in China's public health level, and inter-regional differences are the main source.

In addition, according to the first law of geography, everything within a spatial range is related, and if the distance is different, the interaction between the two also varies significantly. This spatial interaction is also understood as a spatial effect. This spatial effect can be divided into spatial dependence and spatial heterogeneity. Spatial dependence mainly refers to the fact that individuals in space are not independent of each other, but rather interconnected, which is mainly caused by the spillover of factors, technologies, and policies between regions. Spatial heterogeneity is due to different geographical locations and natural resource conditions, leading to certain differences between regions, such as coastal and inland, southern and northern, eastern and western, etc. However, existing studies on the evaluation of public health levels often use traditional econometric models, which assume that individuals exist completely independently in space, do not comply with the first law of geography, and there is a certain computational bias. This study starts from the actual situation in China, based on the guiding ideology and strategic goals of the "Healthy China 2030" Plan issued by the Central Committee of the Communist Party of China and the State Council, and draws on the reasonable parts of existing research, attempting to construct a Chinese public health evaluation index system from five dimensions: popularizing healthy life, optimizing health services, improving health insurance, constructing a healthy environment, and developing health industry. This evaluation index includes various levels of economy, society, ecology, and involves the flow of various resource elements, therefore, there is a significant spatial correlation. In addition, the convergence theory of neoclassical economics suggests that under the condition of diminishing marginal utility of capital in various regions, the growth rate of economically underdeveloped regions is higher than that of economically developed regions. With the promotion of technology, this gap continues to decrease over time, and the economic development level of each region is ultimately in a balanced state. Therefore, the convergence theory can also be applied to the development process of public health in China, where low level provinces of public health will gradually narrow the gap with high level provinces of public health under the combined effect of technology and policies, presenting a convergence characteristic. Based on the above analysis, hypothesis 2 of this paper is proposed: there is a significant spatial correlation in China's public health level, and it shows a certain convergence trend over time.

The rationality of research methods is mainly reflected in the following aspects: firstly, the level of public health in China constructed in this study is an evaluation index system that includes five sub dimensions, including healthy living, health services, health insurance, health environment, and health industry, with 30 specific indicators. In order to ensure the objectivity of measurement results, this study abandons subjective research methods such as expert scoring and Analytic Hierarchy Process, The entropy method, which can objectively weight indicators based on the degree of data dispersion, was chosen to ensure the accuracy of the measurement results. Secondly, according to Hypothesis 1, the actual situation of China's economic and social development may lead to certain regional differences in its public health level. However, the Dagum Gini coefficient improves the drawbacks of the coefficient of variation and the Thile index in measuring regional differences. Therefore, this study chooses the Dagum Gini coefficient to measure the regional differences and main sources of China's public health level. Thirdly, exploratory spatial data analysis is a prerequisite for conducting spatial econometric analysis. According to the first law of geography in hypothesis 2, individuals are closely related in space. Therefore, the global Moran index is used to explore their spatial correlation. Finally, based on the convergence law of classical economy and spatial correlation, a spatial econometric model is constructed to explore the convergence characteristics of China's public health level, and different methods are used for verification.

Reviewer 4 Report

Summary

This paper draws on existing research and health-related statistics covering the four regions of China to analyse the state of public health across the nation by assessing the popularization level of healthy life, the optimization level of health services, the improvement level of health insurance, the health state of environment, and the development level of health industry.

Cautionary note

Sophisticated analytical tools are used (entropy method, Dagum Gini coefficient, Kernel density function and spatial econometric model) with which this reviewer is not familiar. This reviewer found the results and analysis difficult to comprehend. It is suggested that the journal refer the paper to a specialist who can assess the use of these analytical tools.

This stated, the paper is impressive in the breadth of its analysis and in deriving clear conclusions and policy suggestions.

Detailed comments

Line 11 Abstract. The study commences with data from 2009 so it covers the 2009 – 2020 period.

Lines 55 – 57 “In order to solve the problems of low quality of medical and health services, imbalance in resource allocation, and significant differences in public health levels.” This is not a sentence – no verb.

Line 70 “and the 7.20 extremely”

Line 80 “evolution law” What is this? evolution of law?

Line 86 “damaged he production” the

Line159-160 “new era of increasingly rich public health connotations.” What does this mean?

Line 229 “Construction level of healthy environment.” Construction means building a physical structure and is not the right term to use here. “Level of healthy environment” is acceptable.  “Development” could be used instead of Construction.

Line 231 “Construction level of healthy environment.” “State of the natural environment” would be preferable.

Line 252 “…ensure the scientificity…” Scientificity is not a word. “scientific accuracy” is preferable.

Line 413 – 415 “From 2009 to 2020, China's provincial public health level was between 0.1699 and 0.3183, which was generally at a low level. However, it can be found that China's provincial public health level increased from 0.1699 in 2009 to 0.3183 in 2020…” This is repetitious of the same figures. Combine. E.g. “China's provincial public health level rose from 0.1699 in 2009 to 0.3183 in 2020, which was generally at a low level. This was an increase of 0.1484, with an annual growth ….”

Line 473 “…new crown epidemic…” This should be corona epidemic or covid-19 epidemic. Also in line 854.

Line 508 Although the Figure 3 images are novel, they are difficult to interpret and gain the message from each. Transferring them to a graph in the traditional way is recommended.

Line 532 The legend for these maps is low level, lower level, medium level and high level. Lower level implies that it is lower than the low level. You need to reverse low and lower levels so that it reads lower level (or lowest level), low level, medium level, high level.

Line 647 – 655. This is a single sentence of 9 lines which needs to be split into 2 or 3 sentences for the reader.

Line 757 cndition – condition

Line 842 – 848 Split this long sentence into two at line 845.

Summary

This paper draws on existing research and health-related statistics covering the four regions of China to analyse the state of public health across the nation by assessing the popularization level of healthy life, the optimization level of health services, the improvement level of health insurance, the health state of environment, and the development level of health industry.

Cautionary note

Sophisticated analytical tools are used (entropy method, Dagum Gini coefficient, Kernel density function and spatial econometric model) with which this reviewer is not familiar. This reviewer found the results and analysis difficult to comprehend. It is suggested that the journal refer the paper to a specialist who can assess the use of these analytical tools.

This stated, the paper is impressive in the breadth of its analysis and in deriving clear conclusions and policy suggestions.

Detailed comments

Line 11 Abstract. The study commences with data from 2009 so it covers the 2009 – 2020 period.

Lines 55 – 57 “In order to solve the problems of low quality of medical and health services, imbalance in resource allocation, and significant differences in public health levels.” This is not a sentence – no verb.

Line 70 “and the 7.20 extremely”

Line 80 “evolution law” What is this? evolution of law?

Line 86 “damaged he production” the

Line159-160 “new era of increasingly rich public health connotations.” What does this mean?

Line 229 “Construction level of healthy environment.” Construction means building a physical structure and is not the right term to use here. “Level of healthy environment” is acceptable.  “Development” could be used instead of Construction.

Line 231 “Construction level of healthy environment.” “State of the natural environment” would be preferable.

Line 252 “…ensure the scientificity…” Scientificity is not a word. “scientific accuracy” is preferable.

Line 413 – 415 “From 2009 to 2020, China's provincial public health level was between 0.1699 and 0.3183, which was generally at a low level. However, it can be found that China's provincial public health level increased from 0.1699 in 2009 to 0.3183 in 2020…” This is repetitious of the same figures. Combine. E.g. “China's provincial public health level rose from 0.1699 in 2009 to 0.3183 in 2020, which was generally at a low level. This was an increase of 0.1484, with an annual growth ….”

Line 473 “…new crown epidemic…” This should be corona epidemic or covid-19 epidemic. Also in line 854.

Line 508 Although the Figure 3 images are novel, they are difficult to interpret and gain the message from each. Transferring them to a graph in the traditional way is recommended.

Line 532 The legend for these maps is low level, lower level, medium level and high level. Lower level implies that it is lower than the low level. You need to reverse low and lower levels so that it reads lower level (or lowest level), low level, medium level, high level.

Line 647 – 655. This is a single sentence of 9 lines which needs to be split into 2 or 3 sentences for the reader.

Line 757 cndition – condition

Line 842 – 848 Split this long sentence into two at line 845.

Author Response

Response to Reviewer 4 Comments

Dear Reviewer,

Thanks for your comments concerning our manuscript entitled “Regional Differences, Dynamic Evolution and Convergence of Public Health Level in China” (ID: healthcare-2347533). According to these suggestions, we have made several modifications in this paper. Our responses to each suggestion for revision are as follows:

Point 1: Line 11 Abstract. The study commences with data from 2009 so it covers the 2009 – 2020 period.

Revision: Thank you for your feedback. We have changed the time of Line 11 Abstract to "2009-2020" based on your feedback and marked it in revision mode in the original manuscript.

Point 2: Lines 55 – 57 “In order to solve the problems of low quality of medical and health services, imbalance in resource allocation, and significant differences in public health levels.” This is not a sentence – no verb.

Revision: Thank you for your careful review. Based on your feedback, we have updated the "In order to solve the problems of low quality of medical and health services, balance in resource allocation, and significant differences in public health levels.” modified to “The Chinese government aims to solve the problems of low quality of medical and health services, balance in resource allocation, and significant differences in public health levels.”

Point 3: Line 70 “and the 7.20 extremely”

Revision: Thank you for your careful review. In order to better understand, we have revised the "and the 7.20 extremely heavy rain form in Zhengzhou in 2021" in Line 70 to "and the extreme rainstorm in Zhengzhou on July 20, 2021" based on your feedback, and have annotated it with revisions in the original manuscript.

Point 4: Line 80 “evolution law” What is this? evolution of law?

Revision: Thank you for your careful review. In order to better understand, we have changed the "evolution law" in Line 80 to "evolution rule" based on your feedback, and marked it with revisions in the original manuscript.

Point 5: Line 86 “damaged he production” the

Revision: Thank you for your careful review. We have changed the "damaged he production" in Line 86 to "damaged the production" and marked it with revisions in the original manuscript.

Point 6: Line 159-160 “new era of increasingly rich public health connotations.” What does this mean?

Revision: Thank you for your careful review. Line 159-160 “new era of increasingly rich public health connotations.” It means that with the development of economy and society, China has begun to enter the stage of pursuing the quality of development, and the level of public health has become an important aspect that people pursue, and the demand for public health also involves health services, health security, healthy environment and health industry, so single indicators and absolute indicators cannot reflect the current development situation. Of course, "New Era" is a new judgment of China's development stage at the 19th National Congress.

Point 7: Line 229 “Construction level of healthy environment.” Construction means building a physical structure and is not the right term to use here. “Level of healthy environment” is acceptable.  “Development” could be used instead of Construction.

Reply: Thank you for your modification suggestions. We have carefully considered your suggestions. On the one hand, "Construction level of health environment" emphasizes human subjective initiative, but if it is changed to "Level of health environment", it cannot be reflected. In addition, if 'Development' is used instead of 'Construction', it will not reflect the construction process of the living environment. Therefore, regarding the modification of this term, we have ultimately decided to follow the original text, and we kindly request the evaluation experts to understand. Thank you again for your careful review.

Point 8: Line 231 “Construction level of healthy environment.” “State of the natural environment” would be preferable.

Reply: Thank you for your modification suggestions, but as Line 231 "Construction level of health environment" includes three aspects: natural environment, living environment, and public safety. If it is changed to 'State of the natural environment', it can only include one aspect of 'natural environment' and cannot reflect the two aspects of living environment and public safety. Therefore, after comprehensive consideration, we have not changed Line 231 "Construction level of healthy environment" to "State of the natural environment". Please understand. Thank you again for your careful review of this article.

Point 9: Line 252 “…ensure the scientificity…” Scientificity is not a word. “scientific accuracy” is preferable.

Revision: Thank you for your careful review. Based on your feedback, we have changed the "scientificity" in Line 252 to "scientific accuracy" and marked it with revisions in the original manuscript.

Point 10: Line 413 – 415 “From 2009 to 2020, China's provincial public health level was between 0.1699 and 0.3183, which was generally at a low level. However, it can be found that China's provincial public health level increased from 0.1699 in 2009 to 0.3183 in 2020…” This is repetitious of the same figures. Combine. E.g. “China's provincial public health level rose from 0.1699 in 2009 to 0.3183 in 2020, which was generally at a low level. This was an increase of 0.1484, with an annual growth ….”

Revision: Thank you for your careful review. Based on your feedback, we have made modifications to the two sentences of Line413-415, removed repetitive content, and marked them in revision mode in the original manuscript. Specific modifications are as follows: From 2009 to 2020, China's provincial public health level was between 0.1699 and 0.3183, which was generally at a low level overall, but it can be found that China's provincial public health level increased by 0.1484 in 12 years, with an average annual growth rate of 7.28%, showing an overall good development trend.

Point 11: Line 473 “…new crown epidemic…” This should be corona epidemic or covid-19 epidemic. Also in line 854.

Revision: Thank you for your careful review. We have carefully reviewed and revised the entire text, and have revised all 'brown epidemic ' in the original manuscript to ‘COVID-19 epidemic’, all of which have been marked in revision mode. The specific modifications involve Line 70, Line 437, Line 477, Line 630, and Line 854.

Point 12: Line 508 Although the Figure 3 images are novel, they are difficult to interpret and gain the message from each. Transferring them to a graph in the traditional way is recommended.

Reply: Thank you for your feedback. In order to convey the information expressed in Figure 3 more clearly and in conjunction with Reviewer 1's comments, and to increase comparability between regions, we have unified the scales of the three sub figures 3 (a), 3 (b), 3 (c), and 3 (d) of Figure 3, and marked them in revision mode in the original manuscript. The specific modifications are shown in the following figure:

Point 13: Line 532 The legend for these maps is low level, lower level, medium level and high level. Lower level implies that it is lower than the low level. You need to reverse low and lower levels so that it reads lower level (or lowest level), low level, medium level, high level.

Revision: Thank you for your careful review. Based on your feedback, Figure 4 has been modified, changing the four levels from "Low Level, Lower Level, Medium Level, and High Level" to "Lowest Level, Low Level, Medium Level, and High Level", and marked in revision mode in the original manuscript. The specific modified map is as follows:

Point 14: Line 647 – 655. This is a single sentence of 9 lines which needs to be split into 2 or 3 sentences for the reader.

Revision: Thank you for your feedback. We have followed your feedback and marked the single sentence of 9 lines which needs to be split into 2 senses in the original manuscript in revision mode. Specific modifications are as follows:

The main peak height of the Kernel density curve of the popularization level of healthy life and the optimization level of health service "first decreases and then rises", and the width is "first wide and then narrow", which means that the absolute difference between the popularization level of healthy life and the optimization level of health services expands first and then narrows during the entire sample period, especially after 2015. This empirical result proves that in recent years, under the guidance of the national strategy, provinces have continuously strengthened health service publicity and medical and health service improvement, and the gap between regions is narrowing.

Point 15: Line 757 cndition – condition

Revision: Thank you for your feedback. We have revised Line 757 "cndition" to "condition" according to your feedback and marked it in revision mode in the original manuscript.

Point 16: Line 842 – 848 Split this long sentence into two at line 845.

Reply: Thank you for your feedback. Based on Reviewer 1's feedback, we have reorganized the conclusions of this article, deleted this sentence, and marked it in revised mode in the original manuscript.
